# Volume Transmission Implements Context Factorization to Target Online Credit Assignment and Enable Compositional Generalization

**Matthew S. Bull** *
Shanahan Family Foundation Fellow
Allen Institute + University of Washington
Seattle, WA 98109
matthew.bull@alleninstitute.org

**Po-Chen Kuo,**
Graduate Program in Neuroscience
University of Washington
Seattle, WA 98109

**Andrew L. Smith,**
Shanahan Family PostBac Fellow
Allen Institute + University of Washington
Seattle, WA 98109

**Michael A. Buice**
Allen Institute + University of Washington
Seattle, WA 98109
michaelbu@alleninstitute.org

## Abstract

The modern connectivist framing of neural computation emphasizes the primacy of synaptic communication at the risk of neglecting the influence of the surrounding neuromodulatory environment — a neuron's 'biophysical context.' Decades of experimental work has established two views of neuromodulatory (NMs) influence: 1) NMs significantly alter circuit dynamics and 2) NMs gate synaptic plasticity, acting as a 'third factor' in learning. Here, we unify these perspectives, proposing that neuromodulation via volume transmission implements a powerful computational principle: context factorization. We derive an endogenously neuromodulated Recurrent Neural Network (e-nmRNN) from a rate reduction of NM release, showing how NM concentrations dynamically factorize network connectivity. This framework reveals how multiplicative NM gating distinctly influences dynamical regimes compared to additive input. Crucially, this context factorization enables targeted online credit assignment: learning rules derived for the e-nmRNN are naturally gated by NM concentrations, localizing updates to relevant contexts. We demonstrate that e-nmRNN dynamics can learn to approximate gradient descent, facilitating rapid in-context adaptation akin to meta-learning. Empirically, e-nmRNNs achieve strong compositional generalization in sequence-to-sequence tasks, outperforming baselines and exhibiting greater hyperparameter robustness. Furthermore, when trained on complex multitasking benchmarks, e-nmRNNs develop emergent properties mirroring biological observations, including modularity, cell-type specialization based on NM release, and distinct neuromodulatory timescales encoding task context. The model's interpretability allows us to reverse engineer these emergent structures. Notably, in reinforcement learning tasks, the e-nmRNN learns to encode context and signals like Reward Prediction Error (RPE) within its neuromodulator dynamics, demonstrating a mechanism for RPE-gated online credit assignment essential for learning how to learn. By bridging biophysical mechanisms with computational principles and empirical validation, our work presents e-nmRNNs as a performant, interpretable model for understanding the computational role of neuromodulation in flexible and compositional learning.

---

*

39th Conference on Neural Information Processing Systems (NeurIPS 2025).

# 1    Introduction

The adaptability of biological neural networks relies on neuromodulation [1, 2, 3, 4]. Beyond rapid synaptic transmission, neuromodulators (NMs) influence computation via slower, diffuse volume transmission [5, 6, 7] shaping a neuron's 'biophysical context.' This influences circuit dynamics [8, 9, 10] and gates synaptic plasticity [4, 11, 12, 13]. We propose that volume transmission implements context factorization [14], a computational principle unifying these roles by defining the moment to moment dynamics as a product of the dynamics in different contexts and a belief about which context is most relevant. We hypothesize neural dynamics decompose into a slowly varying context state (NM concentrations) and faster neuronal activity [15, 16, 17], providing a parsimonious mechanism for dynamic network reconfiguration [18, 19, 20].

We formalize this intuition with an endogenously neuromodulated Recurrent Neural Network (e-nmRNN), derived from a rate reduction of a biophysical model of NM volume transmission. The "endogenous" nature is its defining feature: the neuromodulatory state is generated by the network's own recurrent activity, creating a closed-loop, self-modulating system. This allows the network to learn to generate its own internal context states, enabling it to autonomously infer context from an input stream and adapt accordingly. The resulting model is related through a nonlinear transformation (rate → synaptic drive) to the recent model from Costacurta et al [15]. Direct comparison after transformation reveals two structural changes: 1) Costacurta et al [15] incorporate low-rank modulatation passing through a second nonlinearity and 2) direct coupling of context and neuromodulation. Our work complements this perspective by directly connecting biophysical parameters to dynamic connectivity.

We view this alternative derivation, resulting in a closely related but non-identical form, as strong support for the value of this model family. The e-nmRNN forms a context-factorized hypernetwork [18], where the NM state modulates effective synaptic weights, linking to attention [21], state-space models [22], and dynamic networks [23, 19, 24].

Our central claim: context factorization enables targeted online credit assignment and robust compositional generalization. e-nmRNN learning rules are gated by NM state, directing updates to relevant contexts. NM dynamics can converge to approximate gradient descent on effective weights, facilitating in-context adaptation akin to meta-learning [25, 26, 20]. We conduct two main experiments – one in supervised learning and one in reinforcement learning – to probe the emergence of credit-assignment-like dynamics in meta-learning environments. We also conduct two further explorations on the facilitating role such learning-to-learn plays for succeeding in compositionally structured tasks [27, 28].

Our empirical findings underscore the value of volume transmission implemented context-factorization. We show that e-nmRNNs achieve strong compositional generalization, outperforming baselines. On multitasking benchmarks [27], trained networks show adaptive generalization and emergent structures (modularity, cell specialization, timescale separation) mirroring biology. In addition, the model's interpretability allows for neurobiology-inspired, reverse engineering, confirming context encoding in NM dynamics. Furthermore, we demonstrate the framework's relevance to reinforcement learning, showing how the model learns to encode critical learning signals like Reward Prediction Error within its neuromodulatory state, opening the door to RPE-gated online adaptation.

This work bridges neurobiology and machine learning, offering an interpretable architecture with strong inductive biases for flexible, compositional learning for advancing the two-way conversation within neuro-AI [29].

# 2    Endogenously Neuromodulated RNNs as Context-Factorized Hypernetworks

Here we outline the derivation and connection of these equations to other model classes. For a more in-depth discussion please see Appendix section 1 and section 2 respectively.

## 2.1    Derivation from Modulated LIF Neurons

Starting with Leaky Integrate-and-Fire (LIF) neurons where synaptic efficacy is modulated by a "well-mixed" environment of NMs surrounding all neurons [3, 4]. This well-mixed assumption is a

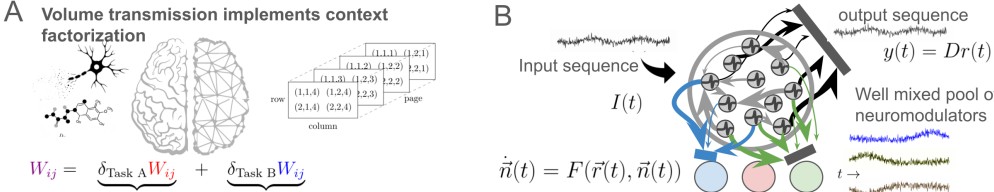

Figure 1: Volume transmission implements context factorization. A) The ambition of this work is to link neurobiological volume transmission of small molecules with a powerful computational primitive: context factorization – the ability to decompose dynamics and context. This approach unifies both targeted online credit assignment with neuronal dynamic changes under different conditions. B) Our tool for connecting substrate with computation is the endogenously neuromodulated RNN characterized by the reciprocal coupling of neuronal rates $\vec{r}(t)$ and neuromodulator concentrations $\vec{n}(t)$ as a principled 'next leading order contribution' to RNN dynamics.

principled abstraction, interpreted as a timescale separation where diffuse NM concentrations establish a stable "context" for faster neural dynamics. We codify this to leading order by its contribution to the effective synaptic strength $W_{ij}^{eff}$. We add a matrix Taylor Expansion around the baseline synaptic weights ($W_{ij}^{o}$) in the form of a learned tensor $T_{ijk}$. This tensor should be interpreted as how the effective coupling between neurons i and j changes in the presence of the kth NM $n_k(t)$:

$$W_{ij}^{eff}(t) = W_{ij}^{o} + \sum_{k=1}^{M} T_{ijk} n_k(t)$$

Importantly, NM concentrations follow their own endogenously generated dynamics when spiking neurons release neuromodulators with each spike. Applying a rate reduction (detailed in Appendix section 1), we obtain the e-nmRNN rate dynamics, where $\phi$ and $\theta$ are nonlinear activation functions (e.g., $\phi$ = tanh and $\theta$=ReLU(tanh))) for the rate and neuromodulator dynamics, respectively:

$$\dot{r}_i = \phi \left( \sum_j W_{ij} r_j + \sum_{j,k} T_{ijk} r_j n_k + \sum_l u_{il} I_l \right)$$

$$\dot{n}_k = \theta \left( \sum_j R_{kj} r_j + \sum_l Z_{kl} n_l \right)$$

These equations link biophysical parameters to trainable RNN parameters $W, T, R, Z, u$. The $Z_{kl}$

Table 1: Parameter and State Variable Definitions for a e-nmRNN

| Parameter | Definition | State | Definition |
|---|---|---|---|
| $W_{ij}$ | Synaptic connectivity $(i, j)$. | $r_i$ | Firing rate, neuron $i$. |
| $T_{ijk}$ | Modulatory effect of $k$th NM on synapse $(i, j)$. | $n_k$ | Concentration $k$th NM. |
| $R_{ik}$ | Release of neuromodulator $k$ by neuron $i$. | | |
| $Z_{kl}$ | Effect of neuromodulator $l$ on $k$. | | |
| $u_{il}$ | Input current to neuron $i$ from stimulus $l$. | | |

matrix models effective interactions between neuromodulatory systems (e.g., release of one NM influencing another), not literal chemical reactions, and often converges during learning to approximately diagonal.

Importantly, we can make explicit connections between the notion of "neuron-type" diversity and the structure of these matrices.

- The sign of the columns of $W_{ij}$ encode Dale's law enforcing the pre-synaptic neurotransmitter to be either excitatory or inhibitory [30].
- The pre-synaptic partners have also been shown to encode cell type [31].

- The rows of the $T$ tensor indicate the sensitivity of the post-synaptic neuron to the presence of a volume transmitted small molecule $n_k(t)$.

- The columns of the $R$ matrix indicate how much of each small molecule a neuron's activity promotes the release of, with a highly selective column indicating specialization. For instance, a unit whose activity primarily drives one modulator could be functionally described as a "dopamine-promoting unit".

- The Z matrix defines the amount of small-molecule to small molecule coupling [7] and often found to be approximately diagonal.

Within the e-nmRNN model class we consider three variants. The first axis of variation is the definition of the decoder. On one hand, we consider the classic trainable decoder with a linear readout matrix $D$, and on the other, we consider a neuromodulated decoder where we put the output decoder subject to the neuromodulatory state:

$$O_i(t) = \sum_j D_{ij} r_j(t) \qquad \text{or} \qquad O_i(t) = \sum_j \sum_k D_{ijk} n_k(t) r_j(t)$$

We call these two options the "fixed decoder" and the "modulated decoder" respectively.

The second primary axis of variation considered in the work is the presence or absence of spatial embedding and Dale's law. Recent work has shown that spatial embedding can improve the convergence on compositional tasks [32], improve interpretability [33], and reduce catastrophic forgetting through the mathematical connection to learning with preferential attachment [34]. This acts as a strong, biologically-inspired regularizer, imposing sparsity and encouraging local processing motifs.

To define our spatially embedded networks, we take inspiration from Waxman random graph generation. We begin by hyper-uniformly placing N neurons into a D dimensional space, $\vec{x}$ (here D=2) [35]. Next, we sample each cell randomly as an excitatory or inhibitory neuron denoted by $\iota_i \in \{0, 1\}$. Combining these two terms we generate our network connectivity with:

$$W_{ij} = W_o (-1)_j^\iota e^{\frac{-|x_i - x_j| + \omega_{ij}}{\ell}}$$

where $W_o$ and $\ell$ are the characteristic magnitude and lengthscale respectively. $\omega_{ij}$ is a trainable parameter whose update rule is simply propagated through the backward pass to find a power-propagation-like learning rule [34] which causes large synapses to learn more quickly and maintains sparsity of neurons which are not connected.

## 2.2 Multiplicative gating of dynamics

As noted by Costacurta et al [15], the nmRNN and this work's close cousin, the e-nmRNN, are special forms of gated RNNs (see also [17]). While an RNN is a universal function approximator, it carries a very different inductive bias than the gated RNN [36]. To build intuition for the ramifications of this form of gating on sequence generation, we study the distinguishable signatures of a rank-2 neural network being modulated across a Bifurcation (e.g. Hopf) via multiplicative gating and additive synaptic drive respectively (see also work from Pellegrino et al [24] on a similarly formulated pitchfork bifurcation).

We find that for a $tanh()$ nonlinearity, these two mechanisms require very different symmetry properties and modes of action (e.g. gain modulation versus operating point shift). Further for a broad class of symmetric nonlinearities, additive synaptic drive from a context neuron is less expressive than differential gain modulation. For a more in-depth discussion see Appendix section 5.

While this intuition is not essential to understand this work, we believe that the interested reader will find this discussion insightful as we study training across various forms of Hopf bifurcations in sections 3, 4, and 5.

## 2.3 Interpretation as a Context-Factorized Hypernetwork

We define *context factorization* as the decomposition of a network's effective connectivity into a fast-varying neural activity and a slow-varying modulatory state (the 'context'; see figure 1A).

Mathematically, the effective synaptic weight from neuron $j$ to $i$ is factorized as:

$$W_{ij}^{\text{eff}}(t) = W_{ij}^0 + \sum_{k=1}^{M} T_{ijk} n_k(t)$$

This allows the low-dimensional context vector $\vec{n}(t)$ to dynamically reconfigure the high-dimensional effective connectivity of the network. This is a context-factorized hypernetwork [15, 19], where the NM network generates parameters for the main RNN. The term $\sum_{j,k} T_{ijk} r_j n_k$ represents tertiary interactions. Connections to SLDS, attention, and other architectures are detailed in Appendix 2.

## 3 Context-Targeted Online Credit Assignment

Context factorization enables online, targeted credit assignment, which is ideally local in time and information [12, 11, 4, 37].

### 3.1 Deriving Learning Rules

Considering gradient descent on a loss $\mathcal{L}$, we derive learning rules by matching terms order-by-order [11, 12]. For $T_{ijk}$, the leading order gradient uses the pre-activation input ("synaptic drive") $A_i(t)$ to neuron $i$'s nonlinearity and the instantaneous error signal $\epsilon_a(t) = y_a(t) - y_a^*(t)$:

$$\nabla_{T_{ijk}} \mathcal{L} \propto \underbrace{n_k(t)}_{\text{context gate}} \underbrace{\epsilon_a(t) D_{ai} \phi'(A_i(t)) r_j(t)}_{\nabla_W \mathcal{L}, \text{ the usual gradient}}$$

This 4-factor rule is gated by $n_k(t)$, ensuring updates are directed to the relevant context [18] by the neuromodulator and the transpose of the decoder matrix. $D$. This rule is a temporally-localized approximation to the full BPTT/RTRL gradient[12]. The recurrent state $\vec{r}(t)$ and, critically, the slow neuromodulatory state $\vec{n}(t)$ act as eligibility traces, carrying the necessary temporal context from the past, making the local update effective for tasks requiring credit assignment over intermediate timescales. Other parameter rules are detailed in the Appendix section 3.

### 3.2 Connection to Meta-Learning and Gradient Approximation

The e-nmRNN dynamics allow $W_{ij}^{eff}(t)$ to change via $\vec{n}(t)$ as:

$$\dot{W}_{ij}^{eff} = T_{ijk} \dot{n}_k(t) \rightarrow T_{ijk} \theta \left( \sum_j R_{kj} r_j + \sum_l Z_{kl} n_l \right)$$

Can $T_{ijk} \dot{n}_k(t)$ learn to approximate $-\eta \nabla_{W_{ij}} \mathcal{L}(t)$? To study this question: we conduct numerical experiments in a simple meta-learning sequence learning task where networks are trained to generate a specific example of the sine family of curves when fed in the amplitude, $A_i$, the frequency, $\omega_i$ and the phase, $\phi_i$ (see Figure 2 A). Our results show that trained e-nmRNN dynamics align with gradient descent on a surrogate network (Figure 2C, D). This alignment (highly statistically significant, see Appendix section 4) implies $\dot{\vec{n}}(t)$ can serve as an error readout, consistent with biological third factor [12]. This capability enables rapid in-context meta-learning [20, 26].

## 4 Compositional Generalization via Sequence Mapping

Compositional generalization is crucial for flexible computation. We use a "dictionary mapping" task: input $I(t)$ and output $y^*(t)$ are linear combinations of basis functions with shared latent coefficients $a_k$ (Figure 3A):

$$I(t) = \sum_{k=1}^{K} a_k \mu_k(t), \quad y^*(t) = \sum_{k=1}^{K} a_k \eta_k(t)$$

The network learns $f_\theta(I(t)) \approx y^*(t)$, inferring $a_k$ to reconstruct $y^*(t)$. As a critical knob to conduct scaling experiments, we note that task complexity scales with $K$. In keeping with the theme of oscillatory signals present broadly in the work, we study a basis set of multi-period oscillatory signals.

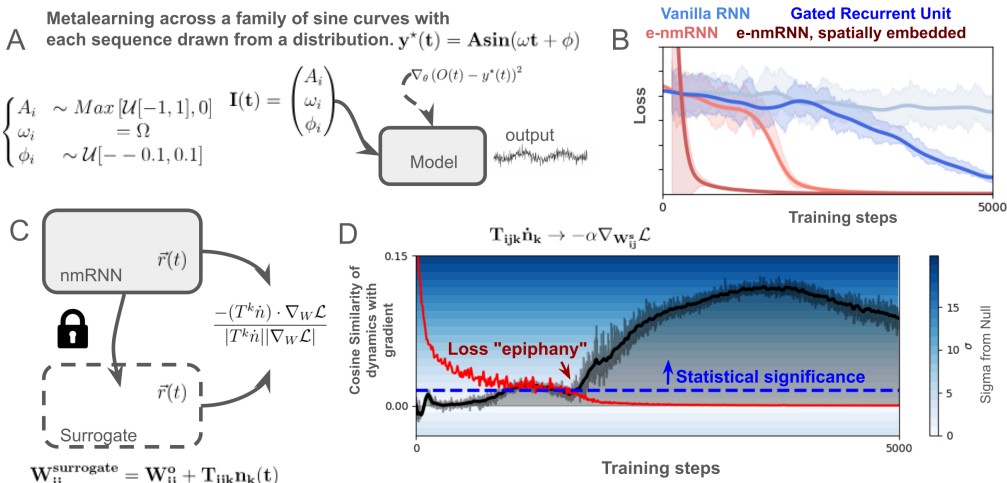

Figure 2: Online and targeted credit assignment arises naturally from context factorization. (A) We define a simple meta-learning task setup where input-output sequence pairs are generated from a family of trajectories (here sine). B) e-nmRNN variants converge rapidly even compared to HP optimized (x100 iterations of TPE) baselines. Spatially embedded networks exploit both power-propagation [34] and context factorization to learn rapidly. C) To measure the alignment of the neuromodulator dynamics of the synaptic weights and the updates that follow the gradient, we develop a surrogate network comparison setup by syncing the nmRNN to an effective RNN. D) Across training, we observe learned alignment between the neuromodulator dynamics and the surrogate model's gradient. After the network undergoes an 'epiphany' in its learning, we measure statistically significant alignment. This suggests that the nmRNN has learned to approximate gradient descent on the parameter updates within its closed loop dynamics (there are no gradient steps executed on the parameters within these validation trials).

We trained e-nmRNNs and baselines (Transformers, HiPPO-LegT [22], Oscillatory RNNs [38], GRU, Vanilla RNN) with matched parameter-count and hyperparameter optimization method (TPE sampling [39]) (Figure 3C). e-nmRNN variants show strong compositional generalization across $K$, outperforming vanilla RNNs and matching GRUs (Figure 3D, E). However, the rich multiplicative dynamics of the e-nmRNN can also make it more prone to overfitting on smaller datasets compared to a GRU, explaining its performance degradation at high task complexity. Spatially embedded (and Dale law enforcing) e-nmRNNs perform robustly overcoming common training challenges [37]. The high coefficient decodability suggests the ability to learn disentangled representations mappable to ground truth that aid in generalization to never seen before combinations. We study this by uniformly sampling the hypercube defined by a space and set aside "quadrants" of combinations of bases for our test set that are not seen during training. It is noteworthy that the nmRNN did suffer from overfitting on small datasets with high complexity limiting its ability to compete with Attention and State-space models (see Appendix section 9.2). Future work will seek biophysical inspiration to directly address this pathology of the model class.

## 5    RPE Encoding and Learning in Reinforcement Learning

The capacity for online, in-context adaptation is particularly critical in dynamic environments requiring reinforcement learning [26]. Previous experimental neuroscience studies have found neural correlates of crucial decision variables in reinforcement learning algorithms. For instance, dopamine activities are found to correlate with reward prediction errors [40] and prefrontal cortex neural activities are found to correlate with action values [41]. We explored the e-nmRNN's capabilities of adaptation in such dynamic environments and examined the emerged neural representations using a dynamic foraging task inspired by neuroscience literature. In a block-switched two-armed bandit task, where reward probabilities invert stochastically (Figure 4A), an agent must learn to track the hidden block state (i.e. which arm is the high probability) and adapt its actions to maximize reward.

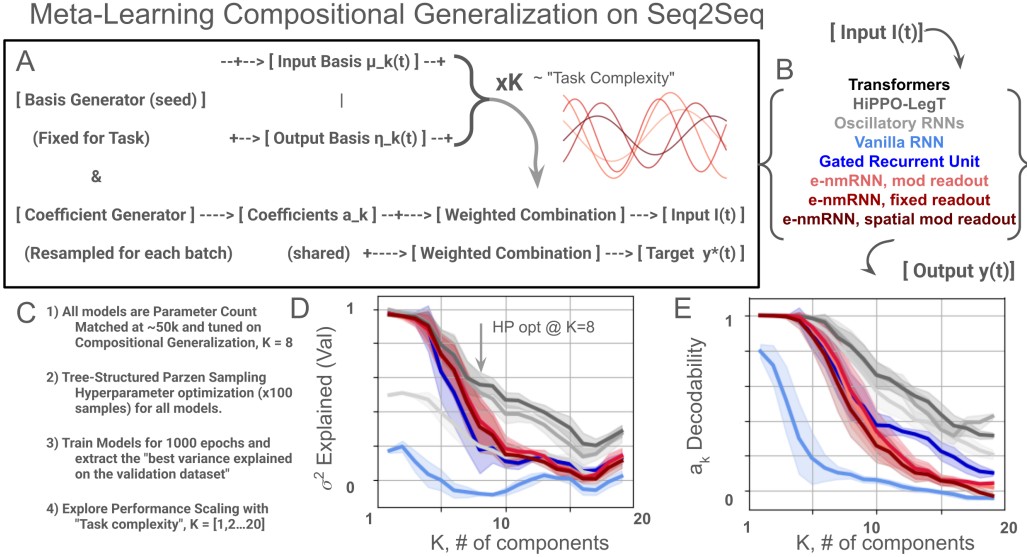

Figure 3: Meta-learning compositional generalization reveals the power of context factorization. A) Compositional task structure. B) Models learn to recognize basis functions, infer coefficients, and generalize to new combinations. C) Training setup with parameter matching and HP optimization is carried out across 4 steps. D) Performance scaling with task complexity ($K$) reveals nmRNN performance exceeding and on par with strong baselines. The model's higher capacity, however, can lead to overfitting on smaller datasets. E) Coefficient decodability with a linear probe shows robust representation encoding in simple environments with a gradual loss as task complexity grows.

Training e-nmRNN agents with Proximal Policy Optimization [42] across vectorized environments of this dynamic foraging task reveals the emergence of adaptive behaviors, including rapid adjustment of choices when the task context changes and anticipatory switching reflecting an internal model of the periodic environment (Figure 4B). Crucially, we find that the neuromodulator dynamics learn to robustly encode the task context (i.e. which arm has high probability) (Figure 4C, D).

Further analysis shows that the neuromodulator state also learns to encode the Reward Prediction Error (RPE) signal derived from a fitted Q-learning model (Figure 4E, G). This is a key finding: it demonstrates how the e-nmRNN can leverage its neuromodulatory system to represent a critical learning signal. Combined with the 4-factor learning rule (Section 3.1), where learning updates are gated by neuromodulator concentrations, this provides a concrete mechanism for RPE-gated online credit assignment. This allows the network to dynamically adjust its effective connectivity based on surprising reward (RPE), facilitating rapid adaptation to changes in the environment and contributing to the model's ability to "learn how to learn" in a reinforcement learning context. Detailed experimental setup and analysis are provided in the Appendix section 6.

## 6 Emergent Biological Signatures and Multitasking

This work echoes the promise of a bold hypothesis for grappling with neuro-biological complexity: optimization of biophysically conforming parameterizations across tasks which prioritize adaptability can provide a principled lens into the paralyzing complexity of the brain [37, 29, 43]. In humble support of this perspective, we report that training spatially embedded, e-nmRNNs (Figure 5A) on the neuroscience-inspired multitasking benchmark from Yang et al. (2019) [27], which covers a wide array of decision-making cognitive tasks (see description in appendix 7), reveals emergent properties consistent with coarse-grained, neuro-biological phenomena. While a one-hot task ID is provided at the start of each trial, the network learns to maintain this information dynamically within its neuromodulatory state. The emergent properties include (Figure 5B):

- **Modularity:** Emergence of functionally specialized modules and hierarchical cell clustering (see Figure 5D). See also complementary findings in Yang et al (2019) [27] and Driscoll et

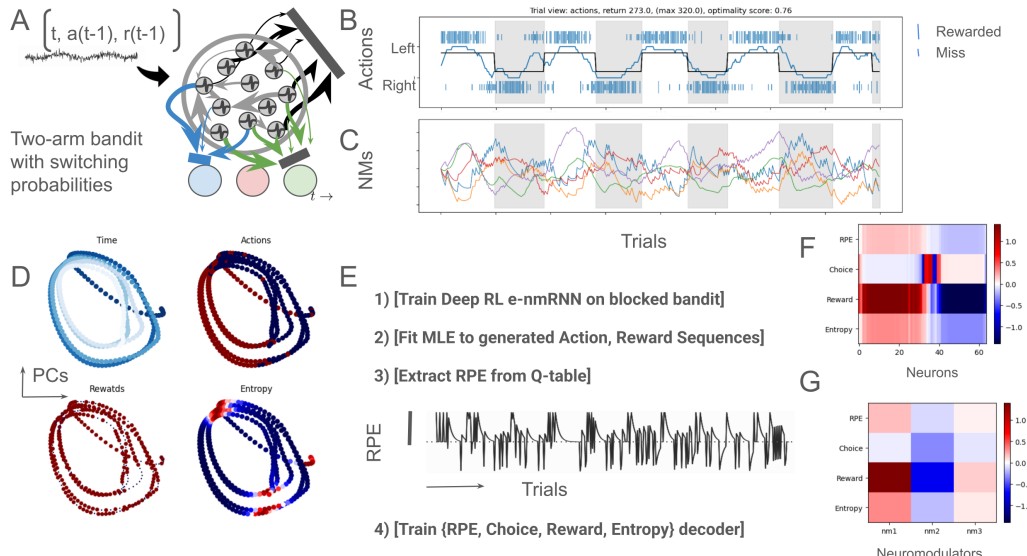

Figure 4: Reward Prediction Error (RPE) encoding emerges from RL training the e-nmRNN. A) Block-switched two-armed bandit task. B) Trained agent actions show adaptive and anticipatory switching. C) Neuromodulator concentration dynamics learn to encode context. D) NM state in PCA space reveals a cyclic trajectory linked to context. E) Example RPE trace from MLE fitted Q-learner. F) Single neuron encoding (via correlation) of behavioral features and RPE. G) Differential NM encoding (via correlation) of behavioral features, including RPE reveals a learned symmetry breaking in NM roles in the dynamics.

al (2024) [28]. Experimentally, this modularity is supported in two ways: 1) the formation of pre-synaptic partners which both form "Sherringtonian circuits" [31, 44] and 2) in modular brain regions [45]. Furthermore, analysis of the trained network reveals that units specializing in releasing a specific neuromodulator form no spatial clusters, mirroring the organization of cell types in the brain (see analysis in Appendix 8.3).

- **Cell-Type Specialization:** Units specialize in releasing specific NMs [10] (Figure 5E). $T_{ijk}$ defines cell types by their sensitivity to specific small molecules. Composed together, we may define the notion of a "dopamine-releasing neuron" as a unit which dominantly promotes the release of a single small molecule via volume transmission (i.e. approximating one-hot columns of the $R$ matrix).

- **Timescale Separation:** NM dynamics learn to evolve on distinct timescales (e.g., 8x factor separating the fastest and the slowest) [10] (Figure 5F).

- **Context Encoding:** NM state $\vec{n}(t)$ learns to represent task context (Figure 5C). A linear decoder trained on NM concentrations can reliably identify the task long after the initial cue has passed, whereas a decoder trained on neural activity cannot. This is a testable prediction for emerging, temporally-resolved neuromodulator imaging technologies.

Through careful, neuroscience inspired reverse engineering of trained networks, these findings support the biological plausibility and interpretability of the e-nmRNN framework. To contextualize these results, we trained a parameter-matched standard RNN on the same benchmark. The spatially-embedded e-nmRNN achieves comparable or superior performance across the 20 tasks, demonstrating that its inductive biases provide tangible value by yielding these emergent, interpretable structures that a standard RNN does not as readily admit. Beyond this support for this class of nmRNNs, we view these results as a small contribution to this greater intellectual landscape attempting to reconcile the lessons of continual optimization in a dynamic environment with the peculiarities of the brain.

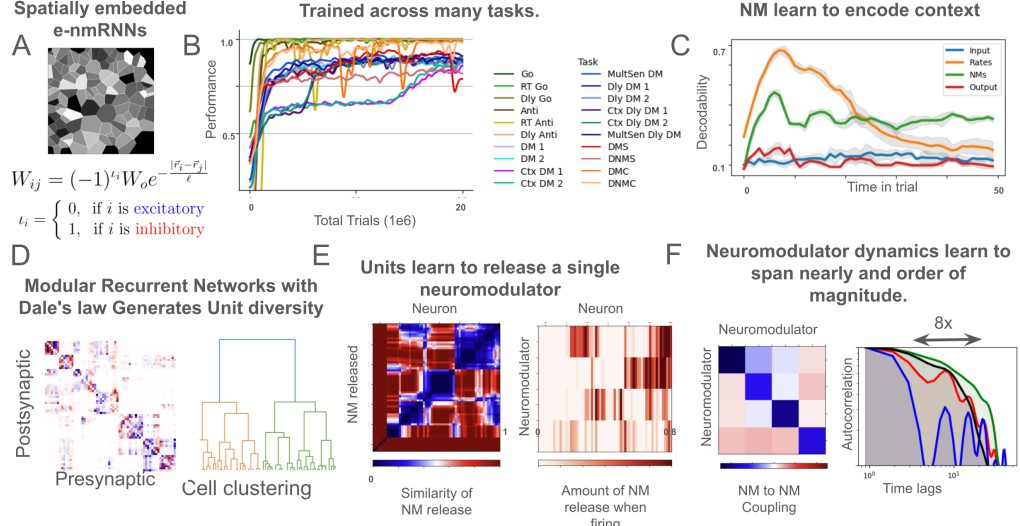

Figure 5: Multitask learning with spatially embedded nmRNNs generates solutions mirroring neural data. (A) Spatial embedding neurons connect most strongly over short distances enforcing sparsity in the trained connectome. (B) Spatially embedded nmRNNs can achieve high performance across all 20 tasks. (C) Training a linear decoder and reporting back the performance on a held-out validation trials, we find that NM concentrations encode context well after trial start, whereas rate encoding of context falls off quickly. (D) Emergent modularity and cell clustering suggests the formation of local circuits within $\sum_t \sum_k T_{ij} n_k$. (E) The resulting $R$ matrix designating small molecule release of the jth neuron reveals unit specialization in NM release (i.e. many neurons release a single small molecule dominantly). (F) Using an autocorrelation metric and its connection of Fluctuation-Dissipation theorem, we find that timescale separation in NM dynamics spans 8x between the fastest and the slowest. This heterogeneity reflects the value of functional specialization.

## 7  Discussion

We interpret neuromodulated RNNs through context factorization via volume transmission, unifying NM roles in dynamics and plasticity. The e-nmRNN, derived from a biophysical model, offers a novel context-factorized hypernetwork architecture. This enables targeted online credit assignment and compositional generalization, supported by empirical results. Emergent biological structures in multitasking networks enhance interpretability. The framework extends to RL, where NM dynamics encode context and RPE, facilitating RPE-gated online adaptation. As a recurrent architecture, the e-nmRNN is inherently sequential during inference and less parallelizable than models like Transformers. This represents a trade-off: we gain powerful inductive biases for online, continuous adaptation at the cost of the massive parallelizability that makes Transformers scalable for large-batch offline processing.

### 7.1  Links between Neuro and AI

This work has significant ramifications for both neurobiology and artificial intelligence. For neurobiology, the e-nmRNN provides a computational hypothesis for the role of volume transmission and neuromodulation in flexible behavior. It suggests that context factorization is a core principle implemented by these mechanisms to manage complexity and enable rapid adaptation. The emergent properties observed in trained e-nmRNNs, such as cell-type specialization based on release profiles and the encoding of context and RPE in NM dynamics, offer experimentally testable predictions about the organization and function of neuromodulatory systems and computation with diverse neuronal components [2]. The distinct influence of multiplicative NM gating on network dynamics compared to additive inputs (see appendix section 5) also provides a specific computational signature that could be sought in biological recordings. In particular, the Hopf bifurcation for a rank-2 network demands

differential modulation of the two modes by a single neuro-modulator (i.e. one of the modes needs to be more or oppositely sensitive relative to the other).

Conversely, insights from neurobiology, particularly the architecture and function of neuromodulatory systems, provide a powerful source of inspiration for advancing machine learning. The e-nmRNN demonstrates how biologically inspired mechanisms like volume transmission and context factorization can lead to more performant and, crucially, more interpretable AI models. The context-factorized structure offers a form of dynamic, interpretable control over network behavior, where the influence of different "context" signals (represented by NMs) on information processing can be directly analyzed. The emergent modularity and timescale separation suggest principles for designing more efficient and scalable AI architectures capable of handling complex, multi-tasking environments. By incorporating biologically plausible inductive biases, we can develop models with enhanced capabilities for online learning, meta-learning, and compositional generalization, addressing key challenges in current AI.

The concept of "computation with diverse components" is central to both fields. In neurobiology, different cell types and neuromodulators contribute distinct computational roles. In AI, designing systems with specialized, interacting modules is a goal. The e-nmRNN framework explicitly models this by incorporating different neuromodulator types with distinct dynamics and effects, and by showing how units can specialize in releasing specific modulators. This provides a concrete model for exploring how diversity in components contributes to overall system function and adaptability.

### 7.2 Limitations

Our work has two primary limitations. First, the model's rich, multiplicative dynamics make it more prone to data-limited overfitting than simpler RNNs when training data is scarce. Future work will explore biologically-inspired regularization methods to mitigate this. Second, the neuromodulator dynamics have intrinsic, learned timescales; if a task required extremely rapid context switching much faster than these timescales, the modulatory state would be unable to keep pace, leading to performance degradation. This highlights a fundamental trade-off between maintaining a stable context representation and achieving rapid adaptability. Finally, the hyperparameter search was extensive but non-exhaustive.

### 7.3 Broader Impacts

Our work contributes to understanding neural computation by bridging biology and AI. The model's interpretability aids in reverse engineering learned solutions and generating biological hypotheses, offering positive scientific impact with low misuse risk (see checklist).

We envision future value in pursuing the monumental task of translating directly between large comprehensive data sets such as the Allen Brain Cell Atlas [46] and compressing them through thoughtful reinterpretation into analogous $W$, $R$ and $T$ structures which can encode powerful initial conditions encoding strong evolutionary inductive biases for future application of the e-nmRNNs and similar model classes. These efforts promise insightful discussion within NeuroAI.

## 8 Conclusion

By formalizing volume transmission's role in context factorization, we introduce the e-nmRNN and join the growing chorus [16, 47, 15, 13, 17], advocating for the unification of NM influence on dynamics and their role in plasticity gating. This framework enables targeted online credit assignment and robust compositional generalization, yielding emergent structures consistent with the rich complexity of neurobiology. The ability of the model's neuromodulator dynamics to encode critical learning signals like RPE highlights its potential for understanding and implementing "learning how to learn" in biological and artificial systems. This work underscores the power of neuro-AI, where insights from neurobiology inspire novel, interpretable, and performant machine learning architectures, while computational models provide a framework for understanding complex biological phenomena like computation with diverse neuromodulatory components. We view e-nmRNNs as a promising model class that bridges AI and neuroscience, highlighting opportunities to advance our understanding in both domains.

We make our code available at `https://github.com/mstormbull/e-nmRNN_neurips2025`

## Acknowledgements

We gratefully acknowledge funding and community support from the Shanahan Foundation Fellowship (MSB and ALS) and the University of Washington's Computational Neuroscience Center (MSB, PCK, and ALS). We acknowledge compute resources provided by the Allen Institute for Neural Dynamics and the eSciences Institute at the University of Washington. We thank Ulises Pereira-Obilinovic, Matthew Golub, Jacob Sacks, Ryan Raut, Anamika Agarwal, Harsha Gurnani, Uygar Sumbul, Bing Brunton, Eric Shea-Brown, Adrienne Fairhall, Laura Driscoll, Kyle Aitken, Stefan Mihalas, Marton Rosza, Kayvon Daie, Karel Svoboda, Christof Koch, and the entire Allen Institute and University of Washington ecosystem for insightful discussions and encouragement. We also thank the Allen Institute founder, Paul G. Allen, for his vision, encouragement, and support.

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

## Appendix

## 1 Detailed Derivation of e-nmRNN Dynamics

**LIF Neuron Model with Neuromodulation via volume transmission**

To capture neurobiological realism, we augment the Leaky Integrate-and-Fire (LIF) neuron model to include the influence of multiple neuromodulator types. The derivation begins with the assumption of a "well-mixed" neuromodulatory environment. This is a principled abstraction representing a timescale separation, where diffuse neuromodulator concentrations establish a stable, slow-varying "context" for the fast neural dynamics (i.e., $\tau_n \gg \tau_r$). This is biologically motivated by two observations: 1) The integrative nature of NM concentration acts as a low-pass filter on population activity, which in turn modifies the mean population dynamics, a useful tool for creating multi-timescale dynamics. 2) The wide cortical projections of single NM-releasing neurons suggest biology may accelerate the approach to a well-mixed state by increasing release site density. We begin with a network of $N$ LIF neurons. The subthreshold dynamics of the membrane potential $V_i(t)$ for neuron $i$ are given by:

$$C_m \frac{dV_i}{dt} = -g_L(V_i - E_L) + I_{syn,i}(t) + I_{ext,i}(t) \tag{1}$$

where $C_m$ is membrane capacitance, $g_L$ is leak conductance, $E_L$ is leak reversal potential, $I_{syn,i}(t)$ is the total synaptic current, and $I_{ext,i}(t)$ is external input current. A spike is emitted when $V_i(t)$ reaches threshold $V_{th}$, followed by a reset to $V_{reset}$ and a refractory period $\tau_{ref}$.

We introduce $M$ types of neuromodulators with concentrations $n_k(t)$, $k = 1, \ldots, M$.

Each cell will be defined by a cell type $z_i$ which controls the distribution of its parameters.

These parameters are a M-dimensional vector (one number for each of the M neuromodulators) which control:

1. $\vec{I}_n$ the neuromodulator current output with a spike
2. $\tau_r$ the timescale of the release from that single cell
3. $\vec{I}_e$ the eligibility current signaling the sensitivity to each neuromodulator
4. $\vec{\tau}_e$ the decay timescale of the eligibility

We constrain these neuromodulator release dynamics such that that shorter acting neuromodulators have a correspondingly higher release. The idea is that we can normalize the area under the curve rather than have faster decaying NMs having significantly less potency.

These parameters control both how much each cell contributes to the pool of neuromodulators and how sensitive they are to that pool via multiplicative gating.

Every cell now has a larger state, the membrane potential plus the eligibility for each of the neuro-modulatory molecules. There is also a shared state for the whole network interacting through the well-mixed neuromodulatory pool.

**Postsynaptic and Presynaptic Effects on Synaptic Current**

Neuromodulators can affect neuronal excitability (postsynaptic) and neurotransmitter release (presynaptic). We model these combined effects by modulating the effective synaptic strength. The total synaptic current is $I_{syn,i}(t) = \sum_{j=1}^{N} I_{ij}(t)$, where $I_{ij}(t)$ is the current from presynaptic neuron $j$. Let $t_j^f$ be the time of the $f$-th spike from neuron $j$.

$$I_{ij}(t) = \sum_f \alpha(t - t_j^f) \cdot W_{ij}^{eff}(t) \tag{2}$$

Here, $\alpha(t)$ is the postsynaptic current kernel (e.g., $\alpha(t) = \frac{Q_{syn}}{\tau_{syn}} e^{-t/\tau_{syn}}$ for $t \geq 0$), and $W_{ij}^{eff}(t)$ is the effective synaptic strength. We model the neuromodulatory effect on this strength as:

$$W_{ij}^{eff}(t) = W_{ij}^o + \sum_{k=1}^{M} T_{ijk}^{LIF} n_k(t) \tag{3}$$

where $W_{ij}^o$ is the baseline weight and $T_{ijk}^{LIF}$ quantifies the sensitivity of the synapse $j \rightarrow i$ to modulator $k$. The superscript 'LIF' distinguishes this parameter from the final RNN parameter.

Substituting (3) into the expression for $I_{syn,i}(t)$:

$$I_{syn,i}(t) = \sum_{j=1}^{N} \sum_{f} \alpha(t - t_j^f) \left( W_{ij}^o + \sum_{k=1}^{M} T_{ijk}^{LIF} n_k(t) \right) \tag{4}$$

**Neuromodulator Dynamics**

The concentration $n_k(t)$ evolves based on release, degradation/uptake, and interactions. Let $S_j(t) = \sum_f \delta(t - t_j^f)$ be the spike train of neuron $j$. We model the dynamics as:

$$\tau_{n,k} \frac{dn_k}{dt} = -n_k + \sum_{j=1}^{N} (\tau_{n,k} R_{kj}^{LIF}) S_j(t) + \sum_{l=1}^{M} (\tau_{n,k} Z_{kl}^{LIF}) n_l(t) \tag{5}$$

Here, $\tau_{n,k}$ is the time constant for modulator $k$, $R_{kj}^{LIF}$ is the amount of modulator $k$ released per spike from neuron $j$, and $Z_{kl}^{LIF}$ represents linear interaction effects between volume-transmitted, small molecules, including decay modification (via $Z_{kk}^{LIF}$) [7]. The interaction term $Z$ models effective interactions, not literal chemical reactions. It allows for one neuromodulatory system to influence another (e.g., release of one NM promoting or inhibiting the release/clearance of another). In practice, we often find this matrix is learned as approximately diagonal, simplifying to self-decay terms.

**Rate Reduction**

We transition to a rate-based description by averaging over spike times, replacing detailed dynamics with average firing rates $r_i(t)$.

**Approximating Firing Rate**

The average firing rate $r_i(t)$ is approximated as a non-linear function $\phi_r$ (the f-I curve) of the mean total input current $\langle I_{tot,i}(t) \rangle = \langle I_{syn,i}(t) \rangle + \langle I_{ext,i}(t) \rangle$:

$$r_i(t) \approx \phi_r(\langle I_{tot,i}(t) \rangle) \tag{6}$$

**Defining and Training Excitability**

Excitability at the single neuron level is encoded in the nonlinear function $\phi()$ that translates the synaptic current into the resulting firing rate. The use of such nonlinearities is standard in rate-based models to ensure stability and represent the saturating, positive firing rates of biological neurons. These nonlinearities also provide important numerical stability during training across a much wider range of hyperparameters.

One way to conceptualize excitability is in terms of a LIF neuron which modulates its excitability by way of changing the threshold needed to cause a spike, $T$. The Leaky Integrate and Fire neuron can be written in a simplified form:

$$\dot{v} = -kv + I - s(t)v_{reset} \tag{7}$$

and is augmented by the fire part of the relationship with the nonlinear spiking function:

$$s(t) = \begin{cases} 0 & \text{if } v(t-1) < T \\ 1 & \text{if } v(t-1) >= T \end{cases} \tag{8}$$

To find the resulting steady-state firing rate of a LIF neuron for a given driving current, I, we can solve this equation in a piecewise linear fashion solving the the differential equation for the time to reach threshold to find that the firing rate takes the form of:

$$f_{LIF}(I) = -k \left( \ln \left| \frac{kT - I}{kV_{reset} - I} \right| \right)^{-1} \tag{9}$$

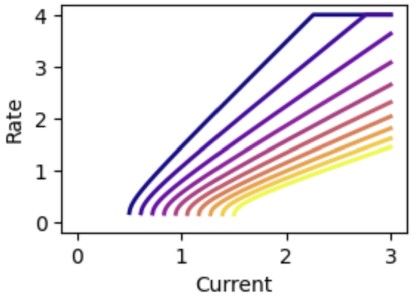

Figure 6: A critical step in the transformation between the LIF neuron and the rate neuron is the shape of the nonlinear mapping between DC input current and output firing rate in the limit of firing rate being fast compared to the dynamics of the input current. We note caution as this assumption only holds for slow directions in the dynamics space. Under this set of assumptions, we derive a closed form equation for the firing rate by completing the piecewise integral of the time to achieve threshold and inverting that to define a firing rate. Notice that this firing rate most closely follows the logistic or the ReLU-Tanh nonlinearity in that it incorporates both a critical threshold for firing and a saturation brought about by the refactory period. Colors in the plot indicate the changing of the threshold of the LIF with direct correspondence to the changes in the slope and the critical current needed to induce firing.

This equation allows us to map the dynamics of a single neuron into a steady state approximation of the firing rate as a function of the input current. This also allows us to study the effect of changing the 'excitability' of the LIF neuron through a change in its threshold to fire. This can be visualized following:

There are three noteworthy aspects to a change in excitability through a change in the threshold.

1. With increasing excitability, the threshold from not firing to firing decreases.
2. With increasing excitability, the slope of the characteristic linear regime increases
3. With increasing excitability, the neuron reaches its maximum firing rate at a smaller input current.

There are then two ways in which we can train the excitability. One, we can directly parameterize our nonlinearity as the LIF rate and use gradient descent to update the LIF threshold value T.

$$\phi() = f_{LIF}(I|T_i) \rightarrow \dot{T} = -\nabla_T \mathcal{L}$$

Or second, parameterize the nonlinearity in terms of two coupled parameters that merge approaches in the literature for training neural excitability:

$$\phi_i() = E_i \text{ReLU}(\tanh() + e_i)$$

where the slope and threshold of the each neuron's nonlinearity is co-modulated by their gradients on the loss function:

$$\dot{E}_i = -\nabla_{E_i} \mathcal{L} \tag{10}$$
$$\dot{e}_i = -\nabla_{e_i} \mathcal{L} \tag{11}$$

The current implementation uses the second form in terms of the ReLU (tanh ()) nonlinearity.

**Approximating Average Synaptic Input**

We replace spike trains $S_j(t)$ with rates $r_j(t)$. From (4), assuming $n_k(t)$ and $r_j(t)$ change slowly relative to $\alpha(t)$:

$$\langle I_{syn,i}(t) \rangle \approx \sum_{j=1}^{N} \langle \sum_f \alpha(t - t_j^f) \rangle \left( W_{ij}^o + \sum_{k=1}^{M} T_{ijk}^{LIF} n_k(t) \right) \tag{12}$$

The average current from neuron $j$ is $\langle \sum_f \alpha(t - t_j^f) \rangle \approx r_j(t) \cdot Q_{tot}$, where $Q_{tot} = \int_0^\infty \alpha(t')dt'$.
Let's define the RNN weights by absorbing $Q_{tot}$: $W_{ij} = Q_{tot}W_{ij}^o$ and $T_{ijk} = Q_{tot}T_{ijk}^{LIF}$.

$$\langle I_{syn,i}(t) \rangle \approx \sum_{j=1}^N r_j(t) \left( W_{ij} + \sum_{k=1}^M T_{ijk}n_k(t) \right) \tag{13}$$

$$= \sum_{j=1}^N W_{ij}r_j(t) + \sum_{j=1}^N \sum_{k=1}^M T_{ijk}r_j(t)n_k(t) \tag{14}$$

**Approximating External Input**

Assume external input $I_{ext,i}(t)$ arises from $L$ signals $I_l(t)$ weighted by $u_{il}$:

$$\langle I_{ext,i}(t) \rangle \approx \sum_{l=1}^L u_{il}I_l(t) \tag{15}$$

**Rate Dynamics Equation**

Standard rate models include a time constant $\tau_r$ for rate dynamics:

$$\tau_r \frac{dr_i}{dt} = -r_i(t) + \phi_r(\langle I_{tot,i}(t) \rangle) \tag{16}$$

Substituting (14) and (15):

$$\tau_r \frac{dr_i}{dt} = -r_i(t) + \phi_r \left( \sum_{j=1}^N W_{ij}r_j(t) + \sum_{j=1}^N \sum_{k=1}^M T_{ijk}r_j(t)n_k(t) + \sum_{l=1}^L u_{il}I_l(t) \right) \tag{17}$$

**Neuromodulator Dynamics in Rate Terms**

Replace $S_j(t)$ with $r_j(t)$ in (5):

$$\tau_{n,k} \frac{dn_k}{dt} = -n_k + \sum_{j=1}^N (\tau_{n,k}R_{kj}^{LIF})r_j(t) + \sum_{l=1}^M (\tau_{n,k}Z_{kl}^{LIF})n_l(t) \tag{18}$$

Define RNN parameters $R_{kj} = \tau_{n,k}R_{kj}^{LIF}$ and effective interaction/decay term $Z_{kl}$ such that the equation becomes (after swapping index $k \to i$ for the equation, and $l \to k$ for the sum):

$$\tau_{n,i} \frac{dn_i}{dt} = \sum_{j=1}^N R_{ij}r_j(t) + \sum_{k=1}^M Z'_{ik}n_k(t) \tag{19}$$

Note that $Z'_{ik}$ combines the original $-n_i$ term and the $Z_{il}^{LIF}$ terms. Explicitly, let $Z''_{ik} = \tau_{n,i}Z_{ik}^{LIF}$. Then the right hand side is $-n_i + \sum_j R_{ij}r_j + \sum_k Z''_{ik}n_k$. Grouping terms involving $n_k$, we get $\sum_j R_{ij}r_j + \sum_k (Z''_{ik} - \delta_{ik})n_k$. So, $Z'_{ik} = Z''_{ik} - \delta_{ik}$.

**Matching the Target Functional Form**

We now compare the derived rate equations (17) and (19) with the target RNN forms in the main text:

$$\dot{r}_i = \varphi \left( \sum_j W_{ij}r_j + \sum_{j,k} T_{ijk}r_j n_k + \sum_l u_{il}I_l \right) \tag{20}$$

$$\dot{n}_i = \varphi_n \left( \sum_j R_{ij}r_j + \sum_k Z_{ik}n_k \right) \tag{21}$$

**Rate Equation ($r$)**

Derived: $\tau_r \dot{r}_i = -r_i + \phi_r(\text{Input}_i)$
Target: $\dot{r}_i = \phi(\text{Input}_i)$
where $\text{Input}_i = \sum_j W_{ij} r_j + \sum_{j,k} T_{ijk} r_j n_k + \sum_l u_{il} I_l$.

The crucial outcome is that the *argument* of the non-linear function $\phi_r$ (or $\phi$) has the structure $W_{ij} r_j + T_{ijk} r_j n_k + u_{il} I_l$, which directly reflects the baseline synaptic drive, the neuromodulated synaptic drive (via effective weight modulation $W_{ij}^{eff}$), and external drive derived from the LIF model.

Obtaining the exact target form $\dot{r}_i = \phi(\text{Input}_i)$ requires interpreting it as a specific modeling choice where the rate dynamics might lack the explicit decay term ($-r_i$) or absorb it and the timescale $\tau_r$ into the definition of $\phi$. The key justification for the target form's structure comes from the LIF derivation of the input terms.

**Neuromodulator Equation ($n$)**

Derived: $\tau_{n,i} \dot{n}_i = \sum_j R_{ij} r_j + \sum_k Z'_{ik} n_k$ (a linear system)
Target: $\dot{n}_i = \phi_n(\sum_j R_{ij} r_j + \sum_k Z_{ik} n_k)$ (a non-linear system)

The derived dynamics for $n_i$ are driven by terms related to release ($\sum_j R_{ij} r_j$) and interactions/decay ($\sum_k Z'_{ik} n_k$). The target form posits a non-linear function $\phi_n$ governing these dynamics, likely representing saturation effects or more complex regulation not captured in the simple linear model (5). The structure of the argument inside $\phi_n$, namely $\sum_j R_{ij} r_j + \sum_k Z_{ik} n_k$, is consistent with the derived driving terms (where $Z_{ik}$ in the target form corresponds conceptually to $Z'_{ik}$ from the derivation, potentially absorbing $\tau_{n,i}$). Using the same functional form $\phi$ as in the rate equation is a further simplifying assumption of the target model.

**Conclusion of Derivation**

Starting from a LIF neuron model incorporating synaptic modulation ($W_{ij}^{eff} = W_{ij}^o + \sum_k T_{ijk}^{LIF} n_k$) and activity-dependent neuromodulator release/dynamics (driven by $R_{kj}^{LIF}, Z_{kl}^{LIF}$), we performed a rate reduction. This yielded:

- An average input current to neuron $i$ with terms corresponding to baseline synapses ($W_{ij} r_j$), modulated synapses ($T_{ijk} r_j n_k$), and external inputs ($u_{il} I_l$).

- A dynamic equation for neuromodulator $n_i$ driven by release ($R_{ij} r_j$) and interactions/decay ($Z'_{ik} n_k$).

The target RNN equations (20) and (21) represent a specific phenomenological model choice for the rate and modulator dynamics ($\dot{r} = \phi(\dots), \dot{n} = \phi_n(\dots)$). The essential link provided by this derivation is the justification for the *structure of the arguments* within the non-linear functions $\phi$ and $\phi_n$. These arguments directly map to the effective inputs and driving forces calculated from the underlying modulated LIF model. The tensors $W, T, u, R, Z$ of the RNN encapsulate parameters from the biophysical level (baseline weights, modulation sensitivities, release rates, interaction constants, etc.), often absorbing scaling factors and timescales during the reduction process. We frame this model as a proof-of-principle showing how even a very low-dimensional modulatory system can powerfully gate high-dimensional dynamics. This system likely works in concert with other biological mechanisms (like thalamocortical gating) to form a richer context space. Our goal is to show how volume transmission provides a powerful, efficient, and interpretable means of broadcasting context.

## 2   Connections to Other Architectures

**Interpretation as a Context-Factorized Hypernetwork**

We define "context factorization" as the decomposition of a network's effective connectivity into a fast-varying neural activity and a slow-varying modulatory state (the 'context'). Mathematically, the

effective synaptic weight from neuron $j$ to $i$ is factorized as:

$$W_{ij}^{\text{eff}}(t) = W_{ij}^0 + \sum_{k=1}^{M} \underbrace{T_{ijk}}_{\text{"dynamics"}} \underbrace{n_k(t)}_{\text{"context"}}$$

This allows the low-dimensional context vector $\vec{n}(t)$ to dynamically reconfigure the high-dimensional effective connectivity of the network. We use the word "factorization" to evoke the mathematical method of decomposing a function into a product of components—here, the components are "context" (encoded in neuromodulator concentrations) and "dynamics" (encoded in the resulting synaptic connectivity patterns).

The derived nmRNN equations (Eq. 20, 21) embody this concept of context factorization[[14, 18]. The effective connectivity $W_{ij}^{eff}(t) = W_{ij} + \sum_k T_{ijk} n_k(t)$ dynamically changes based on the neuromodulator state $\vec{n}(t)$. This state $\vec{n}(t)$, evolving on potentially slower timescales (Eq. 21)[15], acts as the 'context' that modulates the faster neuronal dynamics governed by $\vec{r}(t)$ (Eq. 20)[19].

This structure is equivalent to a specific type of hypernetwork[18]. The neuromodulator network (Eq. 21) acts as the hypernetwork, dynamically generating (part of) the parameters ($T_{ijk}n_k$) for the main RNN (Eq. 20). Both the main network and the hypernetwork have recurrent dynamics[23]. A defining feature of our e-nmRNN is its "endogenous" nature: the neuromodulatory state $\vec{n}(t)$ is generated by the network's own recurrent activity $\vec{r}(t)$ through the learned projection matrix $R$. This creates a closed-loop, self-modulating system. The advantage of this endogenous approach is that it allows the network to learn to generate its own internal context states directly from its population dynamics, enabling it to autonomously infer the current context from an input stream and adapt accordingly. The term $T_{ijk}r_j n_k$ represents tertiary interactions arising from a low-rank approximation to matrix Taylor expansion of the dynamics.

This perspective connects nmRNNs to other dynamic network architectures:

- **Switching Linear Dynamical Systems (SLDS):** Neuromodulation $\vec{n}(t)$ can be seen as a continuous version of the discrete latent state $Z(t)$ in switching Linear Dynamical Systems, smoothly interpolating between different dynamical regimes instead of abruptly switching[48].

- **Fast Weights / Attention:** The dynamic modulation $T_{ijk}n_k(t)$ resembles fast weight programmers or attention mechanisms, where temporal context dynamically gates or modifies information flow from input to output[21, 20].

- **Hypernetworks:** The dynamic modulation $T_{ijk}n_k(t)$ can be viewed as a linear readout hypernetwork which uses a context module to generate the weights of the core network. Here, we note that the nmRNN adopts both recurrent dynamics and the view that the well-mixed neuromodulatory field is governed by volume transmission and therefore neuronal firing [18, 19].

- **High Dimensional Taylor Expansion:** Neuromodulation via volume transmission can be viewed as the next leading order contribution of a Taylor expansion. Many-body physical systems with complex interactions rules can be expanded as:

$$\dot{x}_i = f(\vec{x}) \approx \underbrace{\sum_j A_{ij} x_j}_{\text{Binary Interactions}} + \underbrace{\sum_m \sum_n B_{imn} x_m x_n}_{\text{Three-body interactions}} + \mathcal{O}(\text{Higher order})$$

  . The volume transmission equations can be viewed as a low-rank representation of these three-body interactions proceeding the firing rate nonlinearity and thus represent the next leading order dynamics of the RNN.

- **Koopman operator theory (with a polynomial basis):** In the absence of the nonlinearity, this Taylor expansion can be viewed as a Truncated Koopman linear operator acting over a polynomial dictionary[49]. A valuable way forward may leverage this powerful toolkit for fitting nonlinear dynamical systems.

- **Continuous logic gates:** Gating can be viewed as the continuous "AND" gate where the corners of the space recreate the truth table for discrete input. Through combination of careful construction of the A (rank 2- tensor) and the B (rank 3 tensor), we represent "XOR" gates as well forming the basis of discrete bit universal computation[50]

We advocate for this model as a tool not only to traverse levels of understanding in neurobiology, but also as a scaffold to formally connect diverse approaches for learning in dynamical systems. As a recurrent architecture, the e-nmRNN is inherently sequential during inference, which makes it less parallelizable than models like Transformers. This represents a trade-off: we gain powerful inductive biases for online, continuous adaptation within a single data stream at the cost of the massive parallelizability that makes Transformers so scalable for large-batch offline processing. We position the e-nmRNN as an architecture well-suited for applications like real-time control, agent-based learning, and brain-computer interfaces, where online processing is paramount.

**Connection to Costacurta et al., NeurIPS 2024**

Despite differences in our respective derivations, we observe striking conceptual and experimental similarities to the valuable contribution by Costacurta et al. [15], which also explores the landscape of neuromodulation in recurrent neural networks. To clarify the precise connections and highlight key distinctions, we present a direct comparison of the final functional forms between the two model classes in Figure 7.

Figure 7: A direct comparison with Costacurta et al. reveals both powerful similarities and key architectural differences. The most important distinctions between the two model classes include the low-rank nature of synaptic variability explored in their work, and the absence of direct access to a context signal for our NM dynamics, which enforces an endogenous, closed-system interpretation of these dynamics.

We view our work as highly complementary, yet distinctly advancing the field through several orthogonal contributions:

- **Bio-inspired Derivation**: Our model emerges from a detailed rate reduction of Lif-neuron dynamics incorporating neuromodulation via volume transmission, providing a direct link to established biophysical principles.

- **Novel Learning Rules**: We expand on the model class by deriving normative leading-order learning rules that explicitly link dynamics modification with context-targeted credit assignment, a crucial step for understanding how such systems might learn.

- **Approximation of Gradient Descent**: Crucially, we demonstrate that our e-nmRNNs can learn to approximate gradient descent and robustly encode vital learning signals such as Reward Prediction Error (RPE), providing a computational mechanism for online credit assignment in biological and artificial systems.

- **Dynamic Generalization Tasks**: Our work explores performance on neuroscience-inspired compositional task sets and dynamic generalization tasks, which are distinct in both their formal structure and biological inspiration, showcasing the model's adaptive capabilities.

- **Strong Baseline Comparison**: We conduct rigorous comparisons against strong baselines, systematically assessing the e-nmRNN's capacity and compositional generalization performance.

- **Reverse Engineering Biological Specialization**: The inherent interpretability of the e-nmRNN allows us to reverse engineer cell-type-specific neuromodulator dynamics and emergent cell specialization, offering a novel computational lens for probing neural computation across complex task suites like Yang et al. (2019).

We hope the reader will agree that these contributions are deeply complementary and both serve to advance an important conversation in neuro-AI with their own strengths.

## 3   Detailed Learning Rule Derivations

If we assume that it will look like a local version of gradient descent (in synapse and in time), we can find the gradient of the dynamics that will govern the normative leading order learning rules[11, 37, 12].

Let's define the task in terms of a simple MSE from a target sequence. This target sequence can be represented as a multiple timescales signal via the Laplace transform:

$$y^\star(t) = \int dz a(z) e^{zt}$$

This makes our loss function equal to:

$$\mathcal{L} = \sum_t (y^\star(t) - Dr(t))^2$$

where $D$ is the linear readout matrix that maps neural activity $\vec{r}(t)$ to the output $y(t)$.

We can expand this to timesteps required to percolate information through the network by expanding:

$$r(t) = r(t-1) + dt * \dot{r}(t-1)$$

and

$$n(t) = n(t-1) + dt * \dot{n}(t-1)$$

The first step is writing the error function as:

$$\mathcal{L} = \sum_t \epsilon(t)^2$$

where we assume that $W_{ij}^0 = 0$.

$$\epsilon(t) = y^\star(t) - D\left(r(t-1) + dt\left[-r(t-1) + \phi\left(\sum_j(\sum_k T_{ij}^k n_k(t-1))r_j(t-1) + UI(t-1)\right)\right]\right)$$

Here, $\epsilon(t)$ is the instantaneous error signal, and $\phi$ is the nonlinear activation function for the rate dynamics (e.g., tanh or ReLU-Tanh).

So then we need to plug in the n(t) as:

$$\epsilon(t) = y^\star(t) - D\Bigg(r(t-1) + dt\bigg[-r(t-1) + \phi\bigg(\sum_j\bigg(\sum_k T_{ij}^k\big(n_k(t-1)$$

$$+ dt\big(-n(t-1) + \theta(Rr(t-1) + Zn(t-1))\big)\big)\bigg)r_j$$

$$+ UI(t-1)\bigg)\bigg]\Bigg)$$

where $\theta$ is the nonlinear activation function for the neuromodulator dynamics.

We can expand this out to find:

$$\epsilon(t) = y^\star(t) - Dr(t-1)$$

$$+ dt \left( -r(t-1) + \phi \left( \sum_j (\sum_k T_{ij}^k n_k(t-1)) r_j(t-1) \right) \right)$$

$$+ dt^2 \left( -n(t-1) + \theta(Rr(t-1) + Zn(t-1) + \phi(UI(t-2) + -r(t-2) + \sum_k T^k n_k(t-2)r(t-2))) \right)$$

By plugging these into our dynamics, we can find that our error can be written as an expansion of dt. Local in time calculations and synapse follow the criteria of being to leading order in $\mathcal{O}(dt^2)$. This then allows us to truncate this expansion at leading order.

If we expand this to leading order in dt, we find:

$$\epsilon(t) \overset{\overbrace{\mathcal{O}(dt)}}{\approx} y^\star(t) - Dr(t-1) + dt \left( -r(t-1) + \phi \left( \sum_j (\sum_k T_{ij}^k n_k(t-1)) r_j(t-1) \right) \right)$$

This normative leading order learning rule will take the form of:

$$\nabla_{T^k} \mathcal{L} \overset{\overbrace{\mathcal{O}(dt)}}{\approx} D^T \epsilon(t) \phi'(A(t)) n_k r(t)$$

where $A(t)$ is the pre-activation input or "synaptic drive" to a neuron's nonlinearity.

Interestingly, the presence of the kth neuromodulator should gate the update of the kth page of the three-body tensor. The gating via this multiplicative factor puts the neuromodulator on equal footing to the error.

Next, the gradient of the input sensitivity takes the form of:

$$\nabla_u \mathcal{L} \overset{\overbrace{\mathcal{O}(dt)}}{\approx} D^T \epsilon(t) \phi'(A(t)) n_k I(t)$$

Again, the kth neuromodulator gates the input learning as well.

Next, we can study the gradient of the neuromodulator dynamics with respect to the error:

$$\nabla_R \mathcal{L} \overset{\overbrace{\mathcal{O}(dt^2)}}{\approx} D^T \epsilon(t) \phi'(A(t)) \theta'(B(t)) r^2(t)$$

and the coupling matrix between neuromodulators to leading order takes the form of:

$$\nabla_Z \mathcal{L} \overset{\overbrace{\mathcal{O}(dt^2)}}{\approx} D^T \epsilon(t) \phi'(A(t)) \theta'(B(t)) n(t) r(t)$$

These leading order learning rules could be sensible hypothesizes for how this joint system would evolve if it was tuned to solve the problem of error minimization. (There is an analogous learning rule for minimizing the RL loss derived from Policy Gradients). This update rule is "context-localized" by $n_k(t)$ and "temporally-localized" by using the instantaneous error signal $\epsilon_i(t)$. While this truncates the full gradient through time, the recurrent state $\vec{r}(t)$ and, critically, the slow neuromodulatory state $\vec{n}(t)$ act as eligibility traces, carrying the necessary temporal context from the past, making the approximation potent enough to solve complex tasks requiring credit assignment over intermediate timescales.

## 4  Statistical Significance of Gradient Alignment

In our analysis, we investigate the relationship between the dynamics carried out by the neuromodulators and the gradient of a surrogate model. A surrogate model allows us to calculate a normative

learning signal, by way of ideal gradient to improve performance on the task. We accomplish this by locking the parameters of the e-nmRNN to a coevolving RNN by setting: $W_{ij}^{eff} = W_{ij} + T_{ijk}\bar{n}_k$ over every window of size T = 100. We then use autodiff to calculate how $\dot{W}_{ij}^{eff} = -\alpha\nabla_W\mathcal{L}$ to best reduce the loss of the surrogate network. For the neuromodulator dynamics to "fall into alignment" with this surrogate model's gradient means that the neuromodulator states evolve in a direction that closely matches the optimal update direction prescribed by the surrogate model or in math:

$$T_{ijk}\dot{n}_k(t) \underbrace{\rightarrow}_{\text{learns to approximate}} -\alpha\nabla_W\mathcal{L}$$

**Significance of Cosine Similarity in High Dimensions**

To understand if the observed alignment is significant, we will study observed cosine similarity of $\sim$ 0.15 between two vectors in a high-dimensional space of $D = 128^2 = 16384$ dimensions. In high-dimensional spaces, the distribution of cosine similarities between two randomly and independently drawn vectors tends to concentrate sharply around zero. Specifically, for vectors whose components are drawn independently from a distribution with zero mean and finite variance, the expected cosine similarity is 0. The variance of the cosine similarity between two random unit vectors in $D$ dimensions is approximately $1/D$.

Our null hypothesis ($H_0$) is that the observed cosine similarity arises from two randomly and independently selected vectors, implying no true underlying alignment. Under this null hypothesis, the distribution of cosine similarities has a mean of 0 and a standard deviation of approximately $\sigma = \sqrt{1/D} = \sqrt{1/16384} \approx 0.0078$. The observed cosine similarity of 0.13 is approximately $\sim 16$ standard deviations away from the expected mean of 0 under the null hypothesis. Such a deviation is extremely improbable under this simple null model. Therefore, the observed and slowly accumulated cosine similarity of 0.15 constitutes a statistically significant alignment between the gradient of the surrogate model and the dynamics carried out by the neuromodulators, providing evidence against the null hypothesis that the vectors are unaligned and randomly oriented in this high-dimensional space. This combined with the learned nature of this alignment suggests numerically, that e-nmRNNs not only can, but do learn to approximate gradient descent through gradient descent similar to that observed in meta-optimizers[25] and transformers[20].

## 5 Dynamical Regimes and Bifurcations

**Traversing Bifurcations: Multiplicative vs. Additive Modulation**

Dynamical systems often exhibit bifurcations, where a small parameter change leads to a qualitative shift in behavior (e.g., from a stable fixed point to oscillations). Context factorization provides a distinct mechanism for navigating these critical transitions compared to traditional input modulation, and these mechanisms leave different measurable signatures.

We can illustrate this difference using a simple 2-neuron RNN model:

$$\frac{dr_1}{dt} = -r_1 + \tanh(n(W_{11}r_1 + W_{12}r_2) + r_3) \tag{22}$$

$$\frac{dr_2}{dt} = -r_2 + \tanh(n(W_{21}r_1 + W_{22}r_2) + r_3) \tag{23}$$

Here, $n$ represents multiplicative context factorization (gain modulation), while $r_3$ represents additive context input (e.g., from a dedicated context neuron). Both can drive bifurcations, but how they do so differs fundamentally.

**Multiplicative Modulation ($n$):** When $r_3 = 0$, the origin $(0, 0)$ is always a fixed point due to the odd symmetry of $\tanh$. Linearizing around the origin reveals eigenvalues $\lambda = (2n - 1) \pm in$ for the specific $W$ matrix used in simulations ($W_{11} = W_{22} = 2, W_{12} = -1, W_{21} = 1$). A Hopf bifurcation occurs when $\text{Re}(\lambda) = 0$, which happens precisely at $n = 1/2$. As $n$ increases through $1/2$, the origin transitions from a stable spiral to an unstable spiral surrounded by a stable limit cycle (oscillation)[24]. Critically, this bifurcation happens *without shifting the fixed point away from the origin*. The bifurcation preserves the system's symmetry around $(0, 0)$.

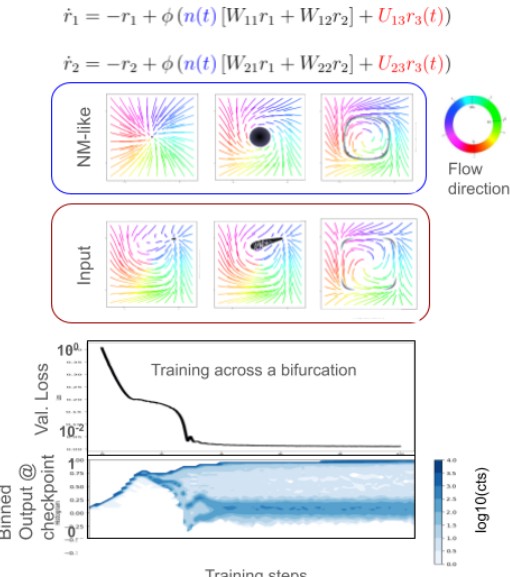

$$\dot{r}_1 = -r_1 + \phi\left(n(t)\left[W_{11}r_1 + W_{12}r_2\right] + U_{13}r_3(t)\right)$$

$$\dot{r}_2 = -r_2 + \phi\left(n(t)\left[W_{21}r_1 + W_{22}r_2\right] + U_{23}r_3(t)\right)$$

Figure 8: Studying the role of gating in the Hopf bifurcation in tuning and in training. A) We study these two 3 dimensional dynamical systems. Black text is present in both. Blue represents context factorization and Red represents the presence of context storing neurons. B) We show the difference of how nm-like and input tuning across the two Bifurcations looks for the tanh() nonlinearity. Flow directions are signaled by the color of the streamline denoted in the circular key to the right. C) We show the evolution of the signal output across training and show that the 'epiphany' moment in the validation loss corresponds to the emergence of the oscillations suggesting the dominant challenge is the formation of the limit cycle.

**Additive Modulation ($r_3$):** When $r_3 \neq 0$, the origin is no longer a fixed point ($\tanh(r_3) \neq 0$). The fixed points $(\bar{r}_1, \bar{r}_2)$ satisfying $\bar{r}_i = \tanh(n(W\bar{\mathbf{r}})_i + r_3)$ shift as $r_3$ changes. Stability is determined by the Jacobian evaluated at these non-zero fixed points:

$$J(\bar{r}_1, \bar{r}_2) = n \begin{pmatrix} (1 - \bar{r}_1^2)W_{11} & (1 - \bar{r}_1^2)W_{12} \\ (1 - \bar{r}_2^2)W_{21} & (1 - \bar{r}_2^2)W_{22} \end{pmatrix} - I$$

Changes in $r_3$ alter $(\bar{r}_1, \bar{r}_2)$ and thus the Jacobian elements, potentially driving eigenvalues across the imaginary axis. However, any bifurcation induced by $r_3$ necessarily occurs at a non-zero operating point $(\bar{r}_1, \bar{r}_2)$ and breaks the symmetry seen when $r_3 = 0$. Depending on the specific path traced by $(\bar{r}_1, \bar{r}_2)$, $r_3$ might induce Hopf or saddle-node bifurcations.

**Distinguishing Mechanisms (Measurable Differences):** These differences provide experimentally testable predictions (visualized in Figure 1B):

1. **Operating Point Shift:** A key distinction is whether the mean activity (average $\bar{r}_1, \bar{r}_2$) shifts significantly as the system crosses a bifurcation. Additive modulation ($r_3$) inherently involves an operating point shift, while multiplicative modulation ($n$ with $r_3 = 0$) can induce bifurcations (like the Hopf bifurcation at the origin) without necessarily changing the mean activity level of that specific fixed point.

2. **Symmetry Preservation/Breaking:** Bifurcations driven by multiplicative gain modulation in a system with appropriate symmetries (like $r_3 = 0$ here) tend to preserve those symmetries (e.g., limit cycle emergence centered around the origin). Additive inputs break this symmetry, leading to asymmetric dynamics relative to the origin.

Observing whether a qualitative change in network dynamics (e.g., onset of oscillations) is accompanied by a shift in the mean firing rates provides a potential signature to distinguish between gain modulation (context factorization) and direct input modulation as mechanisms for context-dependent dynamic switching in biological or artificial networks.

**Bifurcation Analysis of a Rank-2 RNN with Multiplicative and Additive Context Modulation**

We analyze a 2-neuron Recurrent Neural Network (RNN) whose dynamics are given by the following system of coupled ordinary differential equations:

$$\frac{dr_1}{dt} = -r_1 + \phi(n(W_{11}r_1 + W_{12}r_2) + r_3) \tag{24}$$

$$\frac{dr_2}{dt} = -r_2 + \phi(n(W_{21}r_1 + W_{22}r_2) + r_3) \tag{25}$$

where:

- $r_1(t)$ and $r_2(t)$ are the firing rates of the two neurons.

- $\phi(x) = \tanh(x)$ is the non-linear activation function.

- $W = \begin{pmatrix} W_{11} & W_{12} \\ W_{21} & W_{22} \end{pmatrix} = \begin{pmatrix} 2 & -1 \\ 1 & 2 \end{pmatrix}$ is the synaptic weight matrix.

- $n$ is a multiplicative factor representing gain modulation or context factorization. It scales the synaptic inputs before the nonlinearity.

- $r_3$ is an additive factor representing external input or the influence of a 'context neuron'. It shifts the input to the nonlinearity.

Both $n$ and $r_3$ act as control parameters that can drive bifurcations in the system's dynamics. We aim to understand the distinct effects of varying $n$ versus varying $r_3$.

**Fixed Point Analysis**

Fixed points $(\bar{r}_1, \bar{r}_2)$ of the system are solutions to $\frac{dr_1}{dt} = 0$ and $\frac{dr_2}{dt} = 0$. This yields the equations:

$$\bar{r}_1 = \phi(n(W_{11}\bar{r}_1 + W_{12}\bar{r}_2) + r_3) \tag{26}$$
$$\bar{r}_2 = \phi(n(W_{21}\bar{r}_1 + W_{22}\bar{r}_2) + r_3) \tag{27}$$

Let $\mathbf{r} = \begin{pmatrix} r_1 \\ r_2 \end{pmatrix}$ and define the input vector $\mathbf{x} = nW\mathbf{r} + \begin{pmatrix} r_3 \\ r_3 \end{pmatrix}$. Then the fixed point equation is $\bar{\mathbf{r}} = \phi(\mathbf{x}(\bar{\mathbf{r}}))$.

**Symmetry and the Origin**

Consider the case when the additive input $r_3 = 0$. The fixed point equations become:

$$\bar{r}_1 = \tanh(n(W_{11}\bar{r}_1 + W_{12}\bar{r}_2))$$
$$\bar{r}_2 = \tanh(n(W_{21}\bar{r}_1 + W_{22}\bar{r}_2))$$

Since $\tanh(0) = 0$, the origin $(\bar{r}_1, \bar{r}_2) = (0, 0)$ is always a fixed point when $r_3 = 0$, regardless of the value of the multiplicative factor $n$.

Now consider the case when $r_3 \neq 0$. The origin $(0, 0)$ is a fixed point only if:

$$0 = \tanh(n(0) + r_3) = \tanh(r_3)$$
$$0 = \tanh(n(0) + r_3) = \tanh(r_3)$$

This requires $r_3 = 0$. Therefore, the additive factor $r_3$ breaks the symmetry that keeps the origin fixed. An additive input $r_3 \neq 0$ shifts the location of the fixed point(s) away from the origin.

**Measurable Difference 1:** The mean activity (operating point), corresponding to the fixed point location $(\bar{r}_1, \bar{r}_2)$, will generally shift when a bifurcation is induced by changing $r_3$. In contrast, if a bifurcation is induced by changing $n$ while $r_3 = 0$, the origin $(0, 0)$ remains a fixed point, although its stability may change and other fixed points may appear/disappear symmetrically around the origin.

**Stability Analysis and Bifurcations**

To analyze the stability of a fixed point $(\bar{r}_1, \bar{r}_2)$, we linearize the system around it by computing the Jacobian matrix $J$:

$$J = \begin{pmatrix} \frac{\partial \dot{r}_1}{\partial r_1} & \frac{\partial \dot{r}_1}{\partial r_2} \\ \frac{\partial \dot{r}_2}{\partial r_1} & \frac{\partial \dot{r}_2}{\partial r_2} \end{pmatrix}_{(\bar{r}_1, \bar{r}_2)}$$

Let $x_1 = n(W_{11}r_1 + W_{12}r_2) + r_3$ and $x_2 = n(W_{21}r_1 + W_{22}r_2) + r_3$. Recall $\phi'(x) = \text{sech}^2(x) = 1 - \tanh^2(x)$. The partial derivatives are:

$$\frac{\partial \dot{r}_1}{\partial r_1} = -1 + \phi'(x_1) \cdot nW_{11}$$

$$\frac{\partial \dot{r}_1}{\partial r_2} = \phi'(x_1) \cdot nW_{12}$$

$$\frac{\partial \dot{r}_2}{\partial r_1} = \phi'(x_2) \cdot nW_{21}$$

$$\frac{\partial \dot{r}_2}{\partial r_2} = -1 + \phi'(x_2) \cdot nW_{22}$$

At a fixed point $(\bar{r}_1, \bar{r}_2)$, we have $\bar{r}_1 = \phi(\bar{x}_1)$ and $\bar{r}_2 = \phi(\bar{x}_2)$. So, $\phi'(\bar{x}_i) = 1 - \phi^2(\bar{x}_i) = 1 - \bar{r}_i^2$. The Jacobian at the fixed point $(\bar{r}_1, \bar{r}_2)$ is:

$$J(\bar{r}_1, \bar{r}_2) = \begin{pmatrix} -1 + (1 - \bar{r}_1^2)nW_{11} & (1 - \bar{r}_1^2)nW_{12} \\ (1 - \bar{r}_2^2)nW_{21} & -1 + (1 - \bar{r}_2^2)nW_{22} \end{pmatrix}$$

$$J(\bar{r}_1, \bar{r}_2) = n \begin{pmatrix} (1 - \bar{r}_1^2)W_{11} & (1 - \bar{r}_1^2)W_{12} \\ (1 - \bar{r}_2^2)W_{21} & (1 - \bar{r}_2^2)W_{22} \end{pmatrix} - I$$

where $I$ is the identity matrix.

**Stability of the Origin (when $r_3 = 0$)**

When $r_3 = 0$, the origin $(\bar{r}_1, \bar{r}_2) = (0, 0)$ is always a fixed point. Its stability is determined by the Jacobian evaluated at $(0, 0)$:

$$J(0, 0) = n \begin{pmatrix} (1 - 0^2)W_{11} & (1 - 0^2)W_{12} \\ (1 - 0^2)W_{21} & (1 - 0^2)W_{22} \end{pmatrix} - I = nW - I$$

Substituting the values for $W$:

$$J(0, 0) = n \begin{pmatrix} 2 & -1 \\ 1 & 2 \end{pmatrix} - \begin{pmatrix} 1 & 0 \\ 0 & 1 \end{pmatrix} = \begin{pmatrix} 2n - 1 & -n \\ n & 2n - 1 \end{pmatrix}$$

The eigenvalues $\lambda$ are found from the characteristic equation $\det(J(0,0) - \lambda I) = 0$:

$$(2n - 1 - \lambda)^2 - (-n)(n) = 0$$

$$(2n - 1 - \lambda)^2 + n^2 = 0$$

$$\lambda - (2n - 1) = \pm\sqrt{-n^2} = \pm in$$

$$\lambda = (2n - 1) \pm in$$

A bifurcation occurs when the real part of the eigenvalues crosses zero. Here, $\text{Re}(\lambda) = 2n - 1$.

- If $2n - 1 < 0$ (i.e., $n < 1/2$), $\text{Re}(\lambda) < 0$. The origin is a stable fixed point (specifically, a stable spiral since $\text{Im}(\lambda) = \pm n \neq 0$ for $n \neq 0$).

- If $2n - 1 > 0$ (i.e., $n > 1/2$), $\text{Re}(\lambda) > 0$. The origin is an unstable fixed point (an unstable spiral).

- If $2n - 1 = 0$ (i.e., $n = 1/2$), $\text{Re}(\lambda) = 0$. The eigenvalues are purely imaginary ($\lambda = \pm i/2$). This indicates a **Hopf bifurcation** at $n = 1/2$.

As $n$ increases through $1/2$ (with $r_3 = 0$), the stable spiral fixed point at the origin loses stability, giving rise to a stable limit cycle surrounding the now unstable origin. This bifurcation occurs without shifting the location of the fixed point itself (it remains at $(0, 0)$).

**Effect of Additive Input ($r_3 \neq 0$)**

When $r_3 \neq 0$, the origin is no longer a fixed point. The fixed points $(\bar{r}_1, \bar{r}_2)$ must be found numerically or graphically by solving Eqs. 26-27. The Jacobian must be evaluated at these non-zero fixed points:

$$J(\bar{r}_1, \bar{r}_2) = n \begin{pmatrix} (1 - \bar{r}_1^2)W_{11} & (1 - \bar{r}_1^2)W_{12} \\ (1 - \bar{r}_2^2)W_{21} & (1 - \bar{r}_2^2)W_{22} \end{pmatrix} - I$$

As $r_3$ is varied (keeping $n$ fixed, e.g., $n = 1$), the location of the fixed points $(\bar{r}_1, \bar{r}_2)$ will change. The terms $(1 - \bar{r}_i^2)$ in the Jacobian will also change. Bifurcations can occur when the eigenvalues of $J(\bar{r}_1, \bar{r}_2)$ cross the imaginary axis.

Consider the trace and determinant of the Jacobian:

$$\begin{aligned} \text{Tr}(J) &= (-1 + (1 - \bar{r}_1^2)nW_{11}) + (-1 + (1 - \bar{r}_2^2)nW_{22}) \\ &= n((1 - \bar{r}_1^2)W_{11} + (1 - \bar{r}_2^2)W_{22}) - 2 \\ \det(J) &= (-1 + (1 - \bar{r}_1^2)nW_{11})(-1 + (1 - \bar{r}_2^2)nW_{22}) - ((1 - \bar{r}_1^2)nW_{12})((1 - \bar{r}_2^2)nW_{21}) \end{aligned}$$

A Hopf bifurcation occurs when $\text{Tr}(J) = 0$ and $\det(J) > 0$. A saddle-node bifurcation (where fixed points appear or disappear) occurs when $\det(J) = 0$.

Varying $r_3$ changes $(\bar{r}_1, \bar{r}_2)$, which in turn affects both the trace and determinant. This means $r_3$ can potentially drive both Hopf and saddle-node bifurcations, depending on the value of $n$ and the specific path taken by $(\bar{r}_1, \bar{r}_2)$ as $r_3$ changes. Crucially, any bifurcation induced by $r_3$ will be associated with a fixed point $(\bar{r}_1, \bar{r}_2) \neq (0, 0)$.

**Measurable Difference 2:** Bifurcations driven by the multiplicative factor $n$ (with $r_3 = 0$) preserve the symmetry around the origin. For instance, the Hopf bifurcation at $n = 1/2$ happens precisely at the origin. Bifurcations driven by the additive factor $r_3$ occur at non-zero operating points $(\bar{r}_1, \bar{r}_2)$ and generally break the symmetry observed when $r_3 = 0$. The type of bifurcation (Hopf vs. saddle-node) can also differ depending on whether $n$ or $r_3$ is the control parameter.

**Summary of Measurable Differences**

To distinguish between bifurcations driven by multiplicative gain modulation ($n$) versus additive input ($r_3$):

1. **Operating Point / Mean Activity:** Monitor the average values of $r_1$ and $r_2$.
   - Varying $n$ (with $r_3 = 0$): The origin $(0, 0)$ remains a fixed point. Bifurcations (like Hopf at $n = 1/2$) occur centered at the origin. Other fixed points may appear/disappear symmetrically.
   - Varying $r_3$: The fixed point(s) move as $r_3$ changes. Bifurcations occur at non-zero operating points $(\bar{r}_1, \bar{r}_2)$. The mean activity changes significantly across the bifurcation.

2. **Symmetry:** Observe the phase portrait.
   - Varying $n$ (with $r_3 = 0$): The dynamics (fixed points, limit cycles) should exhibit symmetry, e.g., if $(\bar{r}_1, \bar{r}_2)$ is a fixed point, $(-\bar{r}_1, -\bar{r}_2)$ might also be (depending on $W$, though not guaranteed for non-zero fixed points with this specific $W$). The Hopf bifurcation occurs symmetrically around the origin.
   - Varying $r_3$: The symmetry around the origin is broken. Fixed points and limit cycles will generally not be symmetric.

3. **Bifurcation Type at Origin:**
   - Varying $n$ (with $r_3 = 0$): Specifically leads to a Hopf bifurcation at the origin when $n = 1/2$.
   - Varying $r_3$: Cannot induce a bifurcation exactly *at* the origin (as it's not a fixed point for $r_3 \neq 0$). Bifurcations occur elsewhere in the phase space.

By measuring the mean activity levels $(\bar{r}_1, \bar{r}_2)$ and analyzing the symmetry of the dynamics as the control parameter is varied, one can distinguish whether a bifurcation is primarily driven by a multiplicative (gain-like) or additive (input-like) contextual factor.

# 6 Deep Reinforcement Learning Experiments

For the reinforcement learning tasks, we utilized a block-switched two-armed bandit task where reward probabilities invert stochastically (Figure 4A). To address the potential confound of periodic switching, we ran a new experiment where the high-reward arm switched at stochastic, unpredictable intervals. The agent adapted robustly, confirming it learns a genuine model of the task state and is not merely entraining to a fixed frequency. The agent must learn to track the hidden block state (i.e., which arm has high probability) and adapt its actions to maximize reward. Training e-nmRNN agents was performed using Proximal Policy Optimization (PPO), a standard deep reinforcement learning algorithm.

To analyze the RPE encoding, Maximum Likelihood Estimation (MLE) was used to fit a Q-learning model to the generated action and reward sequences from the trained e-nmRNN agent. The RPE signal was then extracted directly from this fitted Q-table. Finally, a single degree-of-freedom linear decoder was trained to map neuromodulator states to various behavioral features, including RPE, choice, reward, and entropy.

Table 2: Key Hyperparameters for Reinforcement Learning Task

| Parameter | Value/Range (PPO/TPE) | Notes |
|---|---|---|
| Batched Environments | 16 | Fixed |
| PPO Learning Rate | $7x10^{-5}$ | Fixed |
| PPO Clip Epsilon | 1 | Fixed |
| Discount Factor ($\gamma$) | 0.95 | Fixed |
| Generalized Advantage Estimation ($\lambda$) | 0.95 | Fixed |
| Sequence Length | 200 | Fixed |
| Hidden Units (RNN/e-nmRNN) | 64 | Parameter-matched |
| Number of NMs | 4 | Fixed |
| NM time constant ($\tau_n$) | 0.1 | Optimized by Hand |
| Activation Function | ReLU-Tanh | Fixed |

The model fitting uses the public github methods from the Allen Institute for Neural Dynamics to fit an MLE to the action sequences of the nmRNN agent.

[language=Python, basicstyle=, breaklines=true] forager = ForagerCollection().$get_p reset_f orager("Bari2019", seed = 42) forager.fit(choice_h istory, reward_h istory, fit_b ounds_o verride$ = $"softmax_i nverse_t emperature" : [0, 100], clamp_p arams$ = $"biasL" : 0, DE_k wargs$ = $dict(workers = 4, disp = True, seed = np.random.default_r ng(42)), k_f old_c ross_v alidation$ = $None, )$

# 7 Multitasking Benchmark Details

Table 4: Summary of Neuroscience-Inspired Cognitive Tasks from Yang et al. (2019).

| Task Name (Abbreviation) | One-Sentence Description (Inspiration) | Task Dynamics (Timing) |
|---|---|---|
| *Go Task Family* | | |
| **Go** (Go) | A pro-response task requiring a response in the direction of a single stimulus (modality 1 or 2) after a go cue. | Stimulus is shown ($T_{stim1} \sim U(500, 1500)$ ms) before the fixation cue goes off. |
| **Reaction-time Go** (RT Go) | A pro-response task requiring a response in the direction of a single stimulus (modality 1 or 2) as soon as it appears. | Response is cued by stimulus onset ($T_{stim1} \sim U(500, 2500)$ ms); fixation cue never goes off. |

Continued on next page

Table 4: Summary of Neuroscience-Inspired Cognitive Tasks from Yang et al. (2019).

| Task Name (Abbreviation) | One-Sentence Description (Inspiration) | Task Dynamics (Timing) |
|---|---|---|
| **Delayed Go** (Dly Go) | A memory-guided saccade task to remember a brief stimulus and respond in its direction after a delay and a go cue. | A brief stimulus is followed by a variable delay ($T_{delay1} \sim U(\{200, 400, 800, 1600\})$ ms). |
| *Anti Task Family* | | |
| **Anti-response** (Anti) | An anti-saccade task requiring a response in the direction *opposite* to a single stimulus (modality 1 or 2) after a go cue. | Same as Go task ($T_{stim1} \sim U(500, 1500)$ ms). |
| **Reaction-time Anti** (RT Anti) | An anti-saccade task requiring a response in the direction *opposite* to a single stimulus (modality 1 or 2) as soon as it appears. | Same as RT Go task ($T_{stim1} \sim U(500, 2500)$ ms). |
| **Delayed Anti** (Dly Anti) | An anti-saccade task to remember a brief stimulus and respond in the *opposite* direction after a delay. | Same as Dly Go task ($T_{delay1} \sim U(\{200, 400, 800, 1600\})$ ms). |
| *Decision Making (DM) Family* | | |
| **Decision Making 1** (DM 1) | A perceptual decision-making task to choose the stronger of two simultaneous stimuli, both in modality 1. | Two stimuli shown simultaneously for $T_{stim1} \sim U(\{400, 800, 1600\})$ ms. |
| **Decision Making 2** (DM 2) | A perceptual decision-making task to choose the stronger of two simultaneous stimuli, both in modality 2. | Two stimuli shown simultaneously for $T_{stim1} \sim U(\{400, 800, 1600\})$ ms. |
| **Context-dependent DM 1** (Ctx DM 1) | A context-dependent computation task to choose the stronger stimulus in modality 1 while *ignoring* modality 2. | Two bi-modal stimuli shown simultaneously for $T_{stim1} \sim U(\{400, 800, 1600\})$ ms. |
| **Context-dependent DM 2** (Ctx DM 2) | A context-dependent computation task to choose the stronger stimulus in modality 2 while *ignoring* modality 1. | Two bi-modal stimuli shown simultaneously for $T_{stim1} \sim U(\{400, 800, 1600\})$ ms. |
| **Multi-sensory DM** (Mult-Sen DM) | A multi-sensory integration task to choose the stimulus with the stronger *combined* strength from both modalities. | Two bi-modal stimuli shown simultaneously for $T_{stim1} \sim U(\{400, 800, 1600\})$ ms. |
| *Delayed Decision Making (Dly DM) Family* | | |
| **Delayed DM 1** (Dly DM 1) | A parametric working memory task to compare the strength of two sequential, brief stimuli in modality 1, separated by a delay. | S1 (300 ms), delay ($T_{delay1} \sim U(\{...1600\})$ ms), then S2 (300 ms) in mod 1. |
| **Delayed DM 2** (Dly DM 2) | A parametric working memory task to compare the strength of two sequential, brief stimuli in modality 2, separated by a delay. | S1 (300 ms), delay ($T_{delay1} \sim U(\{...1600\})$ ms), then S2 (300 ms) in mod 2. |
| **Ctx Dly DM 1** (Ctx Dly DM 1) | A novel task (working memory + gating) to compare sequential bi-modal stimuli based *only* on modality 1. | S1 (300 ms), delay ($T_{delay1} \sim U(\{...1600\})$ ms), then S2 (300 ms); both bi-modal. |

Table 4: Summary of Neuroscience-Inspired Cognitive Tasks from Yang et al. (2019).

| Task Name (Abbreviation) | One-Sentence Description (Inspiration) | Task Dynamics (Timing) |
|---|---|---|
| **Ctx Dly DM 2** (Ctx Dly DM 2) | A novel task (working memory + gating) to compare sequential bi-modal stimuli based *only* on modality 2. | S1 (300 ms), delay ($T_{delay1} \sim U(\{...1600\})$ ms), then S2 (300 ms); both bi-modal. |
| **Multi-sensory Dly DM** (MultSen Dly DM) | A novel task (working memory + integration) to compare the *integrated* strength of two sequential bi-modal stimuli. | S1 (300 ms), delay ($T_{delay1} \sim U(\{...1600\})$ ms), then S2 (300 ms); both bi-modal. |
| *Matching Family* | | |
| **Delayed Match-to-Sample** (DMS) | A visual working memory task to respond to S2 *only if* its direction is the *same* as S1, regardless of modality. | Brief S1, delay ($T_{delay1} \sim U(\{...1600\})$ ms), then brief S2. |
| **Delayed Non-Match-to-Sample** (DNMS) | A visual working memory task to respond to S2 *only if* its direction is *different* from S1, regardless of modality. | Brief S1, delay ($T_{delay1} \sim U(\{...1600\})$ ms), then brief S2. |
| **Delayed Match-to-Category** (DMC) | A visual categorization task to respond to S2 *only if* its direction belongs to the *same category* as S1. | Brief S1, delay ($T_{delay1} \sim U(\{...1600\})$ ms), then brief S2. |
| **Delayed Non-Match-to-Category** (DNMC) | A visual categorization task to respond to S2 *only if* its direction belongs to a *different category* from S1. | Brief S1, delay ($T_{delay1} \sim U(\{...1600\})$ ms), then brief S2. |

We evaluated the e-nmRNN on a suite of 20 neuroscience-inspired multitasking benchmarks from the established suite by Yang et al. (2019), which covers a wide array of decision-making and cognitive tasks. This benchmark suite is designed to probe flexible computation and generalization across diverse cognitive demands. The network is provided a one-hot task ID at the beginning of each trial. The crucial finding is how the network learns to use and maintain this information. The architecture's inductive bias encourages this context to be encoded and dynamically maintained within the neuromodulatory state $\vec{n}(t)$. As shown in Figure 5C, a linear decoder trained on these neuromodulator concentrations can reliably identify the current task long after the initial cue has passed, whereas a decoder trained on the neural activity $\vec{r}(t)$ cannot. This demonstrates that the e-nmRNN learns to represent the task context in its endogenous modulatory state, using it to shape the circuit dynamics appropriately for the task at hand. A parameter-matched standard RNN was also trained on the multitasking benchmark. The results show that our spatially-embedded e-nmRNN achieves comparable or superior performance across the 20 tasks, demonstrating that the inductive biases of our architecture provide tangible value on this complex benchmark. The specific task parameters, input/output structures, and success criteria for each individual task in the suite are detailed in the publicly available code repository (details in Section 10).

# 8 Detailed Analysis of Emergent Biological Structures

Our analysis of trained e-nmRNNs reveals several emergent properties that mirror observations in neurobiology, providing insights into the model's interpretability and biological plausibility.

## 8.1 Modularity and Cell Clustering

Training on multitasking benchmarks promotes the emergence of functionally specialized modules within the e-nmRNN, as well as hierarchical cell clustering (Figure 5D). This modularity is supported by two key observations: 1) the formation of "Sherringtonian circuits" among presynaptic partners[44, 31], and 2) the emergence of distinct, modular brain regions in biological systems. Our clustering

Table 3: Experiment Parameters for the agents shown in the main text.

| Parameter | Value | Parameter | Value |
|---|---|---|---|
| env_name | BlockBandit2ArmCoupledEasy-v0 | policy_use_gae | true |
| exp_label | rl2 | policy_lambda | 0.95 |
| max_episode_steps | 200 | policy_algorithm | a2c |
| max_rollouts_per_task | 1 | policy_critic_loss_coeff | 0.01 |
| seed | 73 | policy_entropy_loss_coeff | 0.3 |
| time_as_state | false | policy_optimizer | adam |
| deterministic_execution | false | policy_eps | 1e-08 |
| results_log_dir | /scratch/ | policy_lr | 0.0007 |
| log_interval | 500 | policy_anneal_lr | false |
| save_interval | 1000 | policy_max_grad_norm | 0.5 |
| save_intermediate_models | false | policy_use_activity_l2_regularization | true |
| eval_interval | 500 | policy_activity_l2_loss_coeff | 100 |
| eval_ids | [] | NMd | true |
| num_eval_envs | 300 | nNM | 4 |
| vis_interval | 500 | state_dim | 1 |
| num_updates | 50000.0 | input_state_dim_for_policy | 0 |
| policy_num_steps_per_update | 200 | action_space_type | Discrete |
| num_processes | 16 | action_dim | 2 |
| deterministic_policy | false | reward_dim | 1 |
| shared_rnn | true | device | cuda |
| layers_before_rnn | [] | | |
| rnn_hidden_dim | 64 | | |
| layers_after_rnn | [] | | |
| rnn_cell_type | vanilla | | |
| action_embed_dim | 0 | | |
| state_embed_dim | 0 | | |
| reward_embed_dim | 0 | | |
| policy_net_activation_function | relu-tanh | | |
| policy_net_initialization_method | normc | | |
| action_pred_type | bernoulli | | |
| hidden_noise_std | 0.002 | | |
| policy_gamma | 0.95 | | |

methods, detailed in Section 10, group neurons based on their connectivity patterns and activity profiles, revealing a clear modular organization that aligns with functional specialization.

## 8.2 Cell-Type Specialization in Neuromodulator Release

We observe a striking specialization where individual units learn to predominantly promote the release of a single type of neuromodulator (Figure 5E). This differentiation is functional and learned, not hardcoded. A unit's activity promotes the release of a specific neuromodulator if the corresponding column in the learned release matrix $R$ becomes selective for that modulator. For instance, if the $j$-th column of $R$, $R_{:j}$, becomes approximately a one-hot vector $[1, 0, 0, ...]^T$, then neuron $j$'s activity $r_j$ primarily drives the dynamics of the first neuromodulator, $n_1$. The functional effects of this neuromodulator are then determined by how its corresponding entries in the learned tensor $T$ gate synaptic transmission throughout the network to solve the given tasks. This emergent property provides a concrete computational model for how diverse neuronal components contribute to overall system function. We also found that units specializing in releasing a specific neuromodulator form no notable spatial clusters, providing new evidence strengthening our claims of emergent, biologically-reminiscent organization.

## 8.3 Spatial Autocorrelation Analysis of Neuromodulator Loadings

To quantify the spatial "mixedness" of the four neuromodulator (NM) loadings across the cell grid, we computed the spatial autocorrelation using Moran's I. This statistic measures the correlation of a variable with itself in spatial terms. The global Moran's I is defined as:

$$I = \frac{N}{S_0} \frac{\sum_{i=1}^{N} \sum_{j=1}^{N} w_{ij}(x_i - \bar{x})(x_j - \bar{x})}{\sum_{i=1}^{N}(x_i - \bar{x})^2}$$

where $N$ is the number of cells (256), $x_i$ is the NM loading for cell $i$, $\bar{x}$ is the mean loading for that NM, and $w_{ij}$ is the spatial weight between cells $i$ and $j$. The spatial weights matrix $W$ was constructed from the Delaunay triangulation of the cell centers ($r$), such that $w_{ij} = 1$ if cells $i$ and $j$ are neighbors (i.e., connected by an edge in the triangulation) and $w_{ij} = 0$ otherwise. $S_0$ is the sum of all weights, $\sum_i \sum_j w_{ij}$. A value of $I > 0$ indicates positive spatial autocorrelation (clustering or low mixedness), $I \approx 0$ indicates no spatial autocorrelation (a random, well-mixed pattern), and $I < 0$ indicates negative spatial autocorrelation (a checkerboard-like, dispersed pattern).

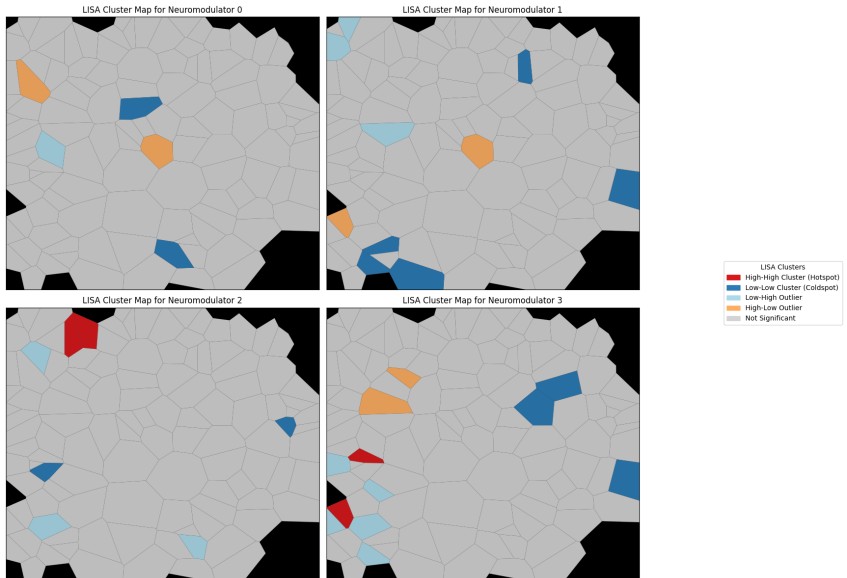

Figure 9: Localizied Spatial Autocorrelation cluster maps for the four neuromodulator loadings (NM 0-3). Global Moran's I analysis did not reveal significant spatial autocorrelation for any neuromodulator (NM0: $p = 0.437$, NM1: $p = 0.216$, NM2: $p = 0.251$, NM3: $p = 0.465$).

To visualize the local spatial structure, we computed the Local Moran's I (LISA) statistic for each cell $i$:

$$I_i = \frac{x_i - \bar{x}}{S_i^2} \sum_j w_{ij}(x_j - \bar{x})$$

where $S_i^2$ is the variance of $x$ (excluding cell $i$). This analysis identifies spatial clusters and outliers, as shown in Figure 9. The map categorizes cells based on their $I_i$ value and its statistical significance (p-value):

- **High-High (Hotspot):** A cell with a high NM loading surrounded by other cells with high loadings.

- **Low-Low (Coldspot):** A cell with a low NM loading surrounded by other cells with low loadings.

- **High-Low / Low-High (Outliers):** A cell that is significantly different from its neighbors.

- **Not Significant:** A cell whose local pattern is not statistically different from a random distribution.

## 8.4 Timescale Separation in Neuromodulator Dynamics

The neuromodulator dynamics within the e-nmRNN spontaneously learn to evolve on distinct, separated timescales (Figure 5F). Our autocorrelation analysis, performed on the time series of individual neuromodulator concentrations, reveals a factor of approximately 8x difference between the fastest and slowest evolving neuromodulators. This heterogeneity in timescales reflects the value of functional specialization, allowing different neuromodulators to encode information at different temporal resolutions, which is consistent with biological observations regarding brain state and behavior. Details of the autocorrelation analysis, including calculation methods and statistical evaluation, are provided in Section 10.

## 8.5 Context Encoding within Neuromodulator Dynamics

Beyond compositional generalization, the neuromodulator state $\vec{n}(t)$ robustly learns to represent task context (Figure 5C). In experiments involving dynamically changing environments, such as the block-switched two-armed bandit task (Figure 4A), the NM concentrations dynamically track the hidden block state (Figure 4C, D). This learned context encoding represents a compelling computational hypothesis for animals trained across multiple tasks or contexts, especially with the advent of emerging, temporally-resolved neuromodulator imaging technologies. This makes a concrete, testable prediction: during multitask learning, ambient NM concentrations in a relevant cortical area should become decodable for task context, a hypothesis that is becoming directly testable with new imaging techniques (e.g., GRAB sensors).

## 8.6 Reward Prediction Error (RPE) Encoding in Reinforcement Learning

In reinforcement learning tasks, specifically the dynamic foraging task, the neuromodulator state not only encodes task context but also learns to represent critical learning signals such as the Reward Prediction Error (RPE) (Figure 4E, G). This is a key finding: it demonstrates how the e-nmRNN can leverage its neuromodulatory system to represent a critical learning signal, consistent with biological observations of dopamine activity correlating with RPE. This RPE encoding, combined with the $n_k(t)$-gated 4-factor learning rule (Section 3.1), provides a concrete mechanism for RPE-gated online credit assignment, facilitating rapid adaptation to changes in the environment and contributing to the network's ability to "learn how to learn" in a reinforcement learning context. This also leads to a testable prediction: in RL settings, some component of the neuromodulatory state space should explicitly encode RPEs, consistent with known roles for dopamine and norepinephrine.

# 9 Experimental Details and Hyperparameters

This section details the experimental setups and hyperparameters used for all reported results, including compositional generalization, multitasking, and reinforcement learning tasks. All experiments were conducted in a cloud environment utilizing AWS, requiring less than 64 GB of RAM and a CUDA-enabled GPU for execution.

## 9.1 Sine Curve Meta-Learning

To assess the model's ability to learn and generalize across a family of structured relationships, we employed a simple meta-learning sequence generation task. Each input-target sequence pair is sampled from a family of sine curves defined by three parameters: amplitude ($A_i$), frequency ($\omega_i$), and phase ($\phi_i$). The target sequence is given by:

$$y^{\star}(t) = A_i \sin\left(\omega_i t + \phi_i\right)$$

The input to the network at each time step is a three-dimensional vector containing these parameters:

$$I(t) = \begin{pmatrix} A_i \\ \omega_i \\ \phi_i \end{pmatrix}$$

Importantly, due to the infinite size of this dataset, we opt to operate in a continual learning regime for this optimization where each sample is both the first and last time the network sees that exact sample. For this reason, we report results not in terms of number of times through the dataset, but instead, number of batched gradient update steps.

The challenge of learning to pair the input parameters with the output sequence should not be underestimated. Indeed, if we sample from the full, uniform distribution of parameters, the networks (all model classes) routinely fail to learn to oscillate. Thus, to focus on operating on both sides of the Hopf Bifurcation, we focused on a fixed distribution on $\omega$ and $\phi$ to sample from $A_i = \max(\mathcal{U}[-1, 1], 0)$.

We began our training with a TPEsampler hyperparameter tuning. Our objective function is the minimization of the held-out loss after 1000 batched gradient update steps.

Next, we applied these locally optimal learning rates to each model class and trained the networks across samples. Batch sizes are held fixed ($B = 16$) across all models to ensure that number of batched gradient updates are compared on equal footing. We find that under these conditions, the e-nmRNN takes many fewer update steps to achieve high performance across this domain and find that the exponentially weighted spatially embedded RNN improve batch update efficiently when paired with the e-nmRNN dynamics. This phenomena, combined with the interpretability advantages of a spatially-embedded network, motivate our continued use throughout this work.

### 9.2 Compositional generalization

This task extends the concept of matrix multiplication to a setting where sequences are mapped to the amplitudes of other sequences. Instead of treating inputs and outputs as static vectors or matrices, we view them as temporal structures, allowing for:

- **Representation Learning**: The input sequences encode a structured representation, and the network must infer a decomposable basis from these representations.
- **Function Approximation**: The network learns to reconstruct an output sequence from a set of basis functions.
- **Compositional Generalization**: If trained correctly, the model should generalize to new sequence compositions beyond its training set.

**Mathematical Formulation of the Task**

We define the **input-output mapping** as:

$$I(t) = \sum_k a_k \mu_k(t)$$

$$y^*(t) = \sum_k a_k \eta_k(t)$$

where:

- $\mu_k(t)$ are the input **basis functions**.
- $\eta_k(t)$ are the output **basis functions**.
- $a_k$ are the unknown **coefficients** that we want the model to infer from the input sequence and use that inference to generate the associated patterns in the corresponding weights.

The model must learn a function $f_\theta$ that maps input sequences to output sequences by implicitly recovering $a_k$:

$$f_\theta(I(t)) \approx y^*(t)$$

**Generalization & Zero-Shot Performance**

To test **generalization**, we evaluate the model on unseen compositions of input sequences:

1. Train the model on a fixed set of basis functions.

2. Introduce new linear combinations of basis functions during testing by holding out combinations of samples from training in the coefficient hypercube. We do this using filtered uniform sampling to split 'quadrants' of the hypercube into a validation set.

3. Measure validation performance on reconstructing previously unseen compositions.

If the model generalizes well, this suggests it has learned an effective representation space that allows for compositional inference.

One signature of learning this representation well is the ability to decode (using a weakly expressive decoder) the actual coefficients of each sequence from the network's hidden units (or nm fields in the case of the e-nmRNN). We add this analysis to our study of the capacity to better understand the failure to generalize with growing dataset complexity.

**Two types of scaling**

One of the great strengths of this simple task set is that we can scale both the complexity of the data set and the amount of data with the only cost being compute time. This allows us to ask how do each of the models selected for baseline characterizations scale with:

1. Scaling with the amount of data available to train the model allows us to test for the "efficiency" of the representation learning

2. Scaling the complexity (by way the number of basis patterns used in the composition, $K$) allows us to measure the practical capacity of the model.

In the work, we explore both of these scaling relationships but only report the results for the capacity due to lack of space. The e-nmRNN's higher capacity makes it more prone to overfitting on this smaller dataset compared to a GRU, which can be seen in the generalization gap between training and validation loss at high task complexity ($K > 15$). A future research topic will investigate effective forms of regularization to avoid this. Future work will explore an analytic explanation of these numerical experiments.

**Reproducibility parameters for compositional generalization**

For the multitasking benchmarks, all models were parameter count matched at approximately 50k trainable parameters and tuned on compositional generalization (K=8). Hyperparameter optimization was performed using Tree-Structured Parzen Sampling with 100 samples for all models. Models were trained for 1000 epochs, and the best variance explained on the validation dataset was extracted for performance evaluation. Performance scaling with "Task complexity" (K ranging from 1 to 20) was explored.

Table 5: Key Hyperparameters for Compositional Generalization Task

| Parameter | Value/Range (TPE) | Notes |
|---|---|---|
| Learning Rate | $[10^{-4}, 10^{-2}]$ | Optimized via TPE sampler |
| Optimizer | Adam | Fixed |
| Batch Size | 16 | Fixed across all models |
| Hidden Units (RNN/GRU) | 256/128 | Parameter-matched |
| Neurons (e-nmRNN) | 128 | Parameter-matched |
| Number of NMs | 4 | Fixed |
| NM time constant ($\tau_n$) | $[0.1, 10]$ | Optimized |
| Weight Regularization (L2) | $[10^{-6}, 10^{-4}]$ | Optimized |
| Activity Regularization (L2) | 0 | Fixed |
| Activation Function | ReLU-Tanh | Fixed |

## 9.3 Multitasking Benchmarks

On the 20 Yang19 tasks, we conducted a single study in the manuscript: what does the solution space high performing e-nmRNN look like? Do we observe biologically aligned phenomena simply by training the nmRNN on a sufficiently rich set of tasks as was seen in [27, 32, 28]

Table 6: Key Hyperparameters for Multitasking Benchmarks

| Parameter | Value/Range (TPE) | Notes |
|---|---|---|
| Learning Rate | $[5 \times 10^{-5}, 5 \times 10^{-3}]$ | Optimized via TPE sampler |
| Optimizer | Adam | Fixed |
| Batch Size | 32 | Fixed |
| Hidden Units | 224 | Parameter-matched ( 50k parameters total) |
| Number of Neurons | 128 | Fixed |
| Number of NMs | 4 | Fixed |
| NM time constant ($\tau_n$) | $[0.01, 10]$ | Optimized |
| Regularization (L2) | $[10^{-7}, 10^{-5}]$ | Optimized |
| Activation Function | ReLU-Tanh | Fixed |
| Dale's Law | Enabled | For spatially embedded networks |
| Spatial Embedding Lengthscale | 10% of domain | For spatially embedded networks |

## 10   Reproducibility Details

To ensure full reproducibility of all experimental results presented in this paper, we will provide access to the codebase, including training scripts, evaluation procedures, and data generation methods. The code will be made available on a GitHub repository post acceptance, and a link for is provided for submission review. Additionally, we plan to provide associated Docker instances hosted on CodeOcean to facilitate ready and consistent reproducibility across different computing environments.

The code repository will include:

- **Source Code:** All Python scripts for defining the e-nmRNN architecture, baseline models (Vanilla RNN, GRU, Transformers, HiPPO-LegT, Oscillatory RNNs), training loops, and evaluation metrics.

- **Environment Configuration:** We provide a `Dockerfile` environment specification detailing all required software libraries and their exact versions (e.g., PyTorch, NumPy, Matplotlib, Optuna, gymnasium).

- **Data Access and Generation:** Scripts for generating the synthetic datasets used in the compositional generalization tasks (sine curves and dictionary mapping). For the multitasking benchmarks and RL tasks, instructions or links to access the generated datasets are provided, along with preprocessing scripts.

- **Hyperparameter Configurations:** Detailed configuration files for the optimal hyperparameters found during the TPE sampler optimization for each experiment, complementing the tables in Section 9.

- **Reproducible Commands:** Clear instructions, including exact command-line arguments, to execute the training and evaluation runs for all results presented in the main text. We present these with saved output as jupyter notebooks for ready inspection.

- **Summary of code structure**: please see below for a detailed summary of code structure focusing on the minimal path toward reproducing notebooks and environment initialization (denoted by a ⋆)

We commit to releasing the assets under a CC-BY 4.0 license. All aspects related to anonymity, as per NeurIPS guidelines, will be strictly adhered to in the publicly released version of the repository.

We make our code available at `https://github.com/mstormbull/e-nmRNN_neurips2025`. Please note that we highlight only the notebooks to run to access the demos but have attempted to provide a complete and blinded version of the code needed to run it.

- `Final_NeurIPS_submission_repo`
    - ⋆ REPRODUCING.md
    - – `Fig2 - Hopf MetaLearning`
        - ⋆ Dockerfile
        - - `v3_refactor` : The full code needed to reproduce gradient alignment

* ⋆ Run the self-contained notebook: `Refactor_CosineSimilarityOverTraining.ipynb`
  * • Folder also includes analysis files of pre-trained logs
- – `Fig3 - Compositional Generalization`
  * ⋆ Environment/Dockerfile
  * ⋆ Hyperparameter tuning can be recreated by running `hyperparameter tuning-Demo.ipynb`
  * ⋆ To train all models in model.py across the task with ascending complexity run [caution: 24 hours execution time]: `GenerateComplexitySweep_MatchedRun.ipynb`
- – `Fig4 - RL Bandit`
  - - `PPO_train_nmRNN`
    * ⋆ environments/Dockerfile
    * • Folder provides access to full repo and some trained example networks.
    * ⋆ To train a new e-nmRNN agent please specify local paths and run [warning execution time 24+ hours]: `TrainNotebook.ipynb`
    * ⋆ To regenerate the dynamics pictures of figure 4 on a new network, please run: `AnalysisOfTrained_nmRNN.ipynb`
  - - `MLE_Fits`
    * ⋆ environments/Dockerfile
    * ⋆ To reproduce the MLE fits and correlation analysis on a previously generated agent sequence please run: `Notebooks/Fit_nmRNN_Agent_Analysis.ipynb`
- – `Fig5 - Multitasking`
  * ⋆ `Yang19MultitaskingAll`
    * ⋆ environment/Dockerfile
    * ⋆ To train a new network please run [execution time 48+ hours] `Demo_SpatialNetwork-Copy3.ipynb`
    * ⋆ To analyze a new network please run `Analysis_spatialNetwork-Alltasks.ipynb`
  - - `AdaptatibilityNeurogym` A repo not used in the paper to study the adaptability of the nmRNN. This repo is in development but welcomes comments, corrections, and contributions.

Note: The code is provided at publication in its current most robust form. Further code revisions are underway to merge the numerous environments while avoiding dependency conflicts.

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
