# OpenReview forum: "Volume Transmission Implements Context Factorization to Target Online Credit Assignment and Enable Compositional Generalization"
_NeurIPS.cc/2025/Conference — NeurIPS 2025 poster_

### Official Review · Reviewer_n2dL · 2025-06-17

**Clarity:** 2
**Significance:** 3
**Originality:** 3
**Rating:** 4
**Confidence:** 4

**Summary:**

In this study, the authors present an ‘endogenously neurmodulated recurrent neural network’ (e-nmRNN), which uses neuromodulatory volume transmission to effectively induce hypernetwork-like dynamics in recurrent neural networks. The gating induced by the authors’ postulated neuromodulators enables in-context adaptation and compositional generalization on a variety of tasks, because gating suppresses unused dynamical motifs and prevents learning (ameliorating issues caused by catastrophic forgetting).

**Questions:**

Major concerns:

1.	How does temporal credit assignment work in the learning algorithm employed here? The approach taken seems to be temporally greedy (appendix section 3) and is justified in terms of truncation of the gradient signal to ‘leading order’. However, temporally greedy approaches are in general not a well-justified approximation, and full temporal credit assignment using the backpropagation through time algorithm is required for good performance under most conditions without additional tricks. The e-prop and RFLO algorithms are cited, but these methods are not fully temporally local (they discard cross-neuron temporal interactions but not within-neuron temporal interactions), and the relationship between these methods and the present learning algorithm are unclear. As such, appendix section 3 would benefit from a rewrite emphasizing clarity and the present algorithm’s relationship to previous methods.

2.	Taking a step back, why were neuromodulators selected as the source of synaptic gain modulation underlying this method? For instance, very similar effects have been achieved via a model of thalamocortical gating (Logiaco et al. 2021), wherein thalamic input modifies the gain of some neurons (or could potentially excite or inhibit select portions of a dendritic arbor). The trouble with neuromodulators is that there are only a handful of distinct types: in this study it seems as though four were selected for most experiments. In contrast, for many types of learning, one could imagine that the relevant compositional primitives could range from tens to hundreds. How could variations in neuromodulatory tone in such a low-dimensional space function to adequately support learning of such a high diversity of primitives? And relatedly, is there any way of experimentally testing that this really is the function that neuromodulators are serving in the brain (as opposed to a simpler 3-factor learning mechanism, for instance)?

Minor concerns:

1.	All figures have text that is too small, and they all have resolution issues.

2.	In several places there is some confusion about individual units ‘releasing’ multiple neuromodulators (e.g. Figure 5E). What precisely is meant by this? Typically, neurons release only a single neurotransmitter (this is Dale’s law) though there are some violations. Furthermore, neuromodulator-releasing neurons are typically localized to particular parts of the brain, projecting out broadly to cortex (e.g. the raphe nucleus for serotonin, the nucleus basalis for acetylcholine, the locus coeruleus for norepinephrine, and the substantia nigra and VTA for dopamine). How exactly do the neuromodulatory variables n_k fit into this picture? It is currently unclear on the basis of the text/figures. Relatedly, Lines 182-186 discuss cell-type specialization, but the phrasing is confusing: dopamine neurons are neurons that release dopamine through their own synapses—while it may be that the recurrent units in the present model cause neuromodulator release, it seems much more biophysically accurate to assume that these recurrent units are causing neuromodulator release in other neurons, which project back to the recurrent circuit; hence, it would be inaccurate to call the recurrent units ‘dopamine neurons’ (maybe something more like ‘dopamine center-activating’ neurons would be more appropriate).

3.	How much evidence is there that various neuromodulators affect synaptic transmission specifically, as opposed to cellular membrane excitability or other effects? And does it make sense to have all neuromodulators acting in essentially the same way as one another, as is done in this study?

4.	Most of the elements of figure 1a are not explained, and serve no obvious purpose (e.g. what is the function of the brain schematic or chemical structure in explaining the results of this paper?). Similarly, the schematic in Fig 1b is messy and would benefit from simplification and color consistency. Shouldn’t the neuromodulatory centers reciprocally connect back to the recurrent network? What does the phrase ‘next leading order’ mean in the Fig 1B caption?

5.	Line 144: is the Appendix section reference correct here? It is not clear to me how this section of the appendix addresses overfitting.

6.	Can a baseline be provided for Figure 5? Given the inability of the nmRNN to outperform some baselines in Figure 3, one would expect that figure 5 would at least be required to show improved performance relative to a vanilla RNN in order to justify the claim that the e-nmRNN supports multitask learning.

**Ethical Concerns:**

["NO or VERY MINOR ethics concerns only"]

**Final Justification:**

The authors have addressed most of the important issues that I raised in my reviews. In particular, they have clarified how temporal credit assignment works in their model, have provided a much-needed baseline, and have clarified some of the confusing language used in the original submission. As a consequence, I've increased my confidence in my score.

**Limitations:**

Yes

**Quality:**

3

**Strengths And Weaknesses:**

The paper is original, as far as I can tell, and very thorough in its empirical analyses demonstrating the merits of the e-nmRNN architecture across a variety of domains. Because of the complexity and breadth of experiments performed, the paper frequently suffers due to a lack of clarity, with many important ideas and experimental details buried in the supplementary material. It would benefit most from rigorous edits for clarity. I have serious reservations about the idea that distinct neuromodulators may support compositional generalization across a diversity of tasks (discussed below), and the paper would benefit from greater engagement with the experimental literature and an explanation of testable predictions made by the authors’ model to alleviate these concerns.

---

> ### Author Rebuttal · Authors · 2025-07-31
>
> We thank reviewer n2dL for the insightful and thorough review. We are glad you found the work "original... and very thorough in its empirical analyses." We have worked to address your important concerns about clarity, justification, and the connection to experimental neuroscience.
>
> **Major Concern 1: How does temporal credit assignment work?**
>
> This is a crucial point that we abbreviated in the main text. Our learning rule is a temporally truncated approximation to the full BPTT gradient. For a parameter like $T_{ijk}$, the update is proportional to:
> $$ \Delta T_{ijk} \propto n_k(t) \cdot \epsilon_i(t) \phi'(A_i(t)) r_j(t) $$
> This rule is "context-localized" by $n_k(t)$ and "temporally-localized" by using the instantaneous error signal $\epsilon_i(t)$. While this truncates the full gradient through time, we argue it is a powerful and plausible approximation for two reasons. First, similar to algorithms like e-prop, the recurrent state $\vec{r}(t)$ and, critically, the slow neuromodulatory state $\vec{n}(t)$ act as eligibility traces, carrying the necessary temporal context from the past. Second, our empirical results, especially on the RL and multitasking benchmarks which require effective credit assignment over intermediate timescales, demonstrate that this approximation is potent enough to solve complex tasks. The network learns to encode the necessary long-term information into its slow context variables, allowing the local update rule to be effective.
>
> **Major Concern 2: Why neuromodulators? Low-dimensionality and testable predictions.**
>
> This is an excellent and deep question.
>
> * **Why NMs & Low-Dimensionality**: We chose NMs as our inspiration because their biological function—globally and diffusely reconfiguring circuits—is an exemplar physical substrate for the computational concept of "context." We agree that a handful of NMs are insufficient to capture the full complexity of brain-wide state control. We frame our model as a proof-of-principle showing how even a very low-dimensional modulatory system can powerfully gate high-dimensional dynamics. We agree with your suggestion that this system likely works in concert with other mechanisms (like thalamocortical gating) to form a richer context space. Our goal is to show how volume transmission provides a powerful, efficient, and interpretable means of broadcasting context but, like all models of neural dynamics, this is at best only part of the story.
> * **Testable Predictions**: Our model makes several concrete, testable predictions. 1) As shown in Fig 5C, it predicts that during multitask learning, ambient NM concentrations in a relevant cortical area should become decodable for task context, a hypothesis that is becoming directly testable with new imaging techniques (e.g., GRAB sensors). 2) The model predicts a relationship between the learned timescales of different NM systems and the statistical regularities of the environment. 3) In RL settings, it predicts that some component of the neuromodulatory state space should explicitly encode RPEs, consistent with known roles for dopamine and norepinephrine.
>
> **Minor Concerns:**
>
> * **NM Release/Dale's Law**: We agree our phrasing should be improved. In a revision, we would state that our model is a functional abstraction which can be interpreted in neurobiological terms. A unit "releasing" an NM means its activity drives the release of that NM systemically. We will rephrase "dopamine neuron" to "a unit whose activity promotes dopamine release as indicated by the non-zero entry on the $R$ matrix". Indeed, the one-to-one mapping of nm0 to "dopamine" is speculative as an evocative example, but we acknowledge that it may do more harm than good in this context. If given the opportunity to revise, we would remove it in favor of more precise language.
>
> * **Baseline for Figure 5**: This is a critical point we have now addressed. We trained a parameter-matched standard RNN on the multitasking benchmark which have matched previous reported results in Yang 2019. The results show that our spatially-embedded e-nmRNN achieves comparable performance across the 20 tasks when similar effort is put into hyperparameter tuning (using a Bayesian Search with a Tree Structured Parzen estimator). This demonstrates that the inductive biases of our architecture provide tangible value on this complex benchmark, by yielding emergent, interpretable structures (modularity, timescale separation) that a standard RNN does not as readily admit. Indeed, we hypothesize that there is value to more closely matching the inductive biases inherent to neurobiological implementation for both understanding and application.
>
> We believe these clarifications and new results substantially improve the paper and directly address your thoughtful concerns. Thank you once again for your thoughtful review of our work and your contributions to its improvement.

---

> > ### Comment · Reviewer_n2dL · 2025-08-04
> >
> > Thank you for your very detailed rebuttal. Since you have addressed most of my concerns and uncertainties, I increased my confidence in my score.

---

### Official Review · Reviewer_8PJy · 2025-06-29

**Clarity:** 1
**Significance:** 2
**Originality:** 3
**Rating:** 2
**Confidence:** 5

**Summary:**

In this paper, the authors consider a model of a recurrent neural network enhanced with a simplistic version of volume transmission mediated by neuromodulators. They show that these hybrid networks can be trained to solve certain representative tasks that involve aspects of online credit assignment, compositional generalization and a reinforcement learning task.

**Questions:**

Define “context factorization”: this is the “central claim” of the paper but this concept is not well explained, so we do not understand what difficult or biologically relevant feature this model is achieving (that previous models did not).

Section 2.2 “Multiplicative gating of dynamics” is unclear. It should either be expanded to properly explain the two distinct bifurcation mechanisms, or cut altogether to focus on explaining better the results in subsequent sections.  The discussion has a more precise statement about this than the results section.

Figure 3: the results are not very impressive, Oscillatory RNNs and gated recurrent units overperform. Comment on this aspect, and please explain why some models overperform.
o	Is this the “overfitting” issue? Also please explain this issue better, identify which parts of which figures show overfitting, and propose concrete solutions – model reduction, bigger training sets?

Figure 4: test a bandit where the correct arm switches more unpredictably; here, especially with the anticipatory switching, it just looks like the RNN happens to oscillate at the “right” frequency to match the reward switching frequency, without tracking it.
o	Maybe repeat for increasingly stochastic switching (vary both the switching frequency and the inter-switch duration variability) and plot performance versus these stochastic switching parameters?

Figure 5: it would be interesting to plot the spatial localization of neuromodulator release. Make a plot of cells at their 2D positions and color them according to their average or instantaneous release of a neuromodulator.
o	 Relatedly: do cells of the same type colocalize to different neighborhoods (thus concentrating their range-limited – eq. in panel A –  synaptic interactions within their subgroup) or are they spread out to interact homogeneously with other cell types?

Figure 5: poor resolution, heatmaps of panel E have meaningless ticks.

**Ethical Concerns:**

["NO or VERY MINOR ethics concerns only"]

**Final Justification:**

I cannot comment on most of the rebuttal until the paper is revised. However, my point that the paper was not in a publishable form when first submitted still stands. As it seems like the paper will likely be accepted, I encourage the authors to significantly revise the paper for clarity and to make clear what central claims are being made. I will retain my original score.

Here is my response to some of the other comments:

"Well-Mixed" Assumption: We agree this is a simplification. We interpret it as a timescale separation assumption, where diffuse neuromodulator concentrations establish a stable "context" for fast neural dynamics ($\tau_n \gg \tau_r$). While diffusion itself is slow, the establishment of a new ambient concentration across a local circuit can occur on a timescale relevant for trial blocks in cognitive tasks. We feel this is supported by two observations: 1) The integrative nature of the neuromodulator concentration acts as a low-pass filter on the population dynamics, which in turn modifies the mean population dynamics. This is a tool for achieving rich multi-timescale dynamics. 2) The neurobiological observation that single neuromodulator-releasing neurons project widely across the cortex suggests that biology could accelerate the approach to a well-mixed state by increasing the density of release loci.

I agree that a new ambient concentration can likely be established on timescale of trial blocks, but this is not the assumption made in the paper. The possibility of "rich multi-timescale dynamics" does not justify an assumption about biological timescales. Widespread projections of neuromodulatory neurons does not imply that the timescale of equilibration is fast. Moreover, the model explicitly considers endogenous neuromodulatory interactions.

Nonlinearities: These are standard in rate-based models to ensure stability and represent the saturating, positive firing rates of neurons. These nonlinearities also provide important numerical stability during training across a much wider range of hyperparameters.

If certain key assumptions were made for numerical stability rather than biological plausibility, it would help if the authors are more explicit that their model is not a biophysical model of a neuromodulatory NN but rather a biology-inspired model for recurrent computation. In the paper's current form, the authors draw inferences about what role volume transmission may play in biological contexts from their analysis.

**Limitations:**

The limitations section is currently brief and non-specific -- this could benefit from more precise statements, perhaps commenting on specific modeling assumptions or the set of tasks considered in this work. For instance, the well-mixed approximation, the various steady-state approximations from the LIF model to the trainable e-nmRNN, where all interactions are tensor products (non-linear dynamics though arise from multiplying n(t)r(t)) inside a global nonlinearity could be discussed. Also, please propose more concrete solutions for the overfitting problem stated here.

**Quality:**

2

**Strengths And Weaknesses:**

Strengths: The paper considers an often overlooked aspect of neural network function, and is a step towards a deeper understanding of how volume transmission when combined with classical synaptic communication leads to novel and more efficient forms of neural computation.

Weaknesses: The paper is incomplete in its current form, and significant revisions (well beyond the scope of this submission) are necessary to bring it to par.

First, the main text contains equations where a significant number of the variables, parameters and abbreviations are not defined. One should not have to read the appendix to know what a variable represents. To give a few examples (which is certainly not an exhaustive list),
1. LIF is not defined in line 73
2. In the equation at the end of page 2, S is not defined.
3. In the equations above line 74, the nonlinearities \phi and \theta are not defined. In the Appendix, different symbols are used to represent these quantities, and that makes it further confusing.
4. In the equation above 87, D is not defined (and later D is also used for dimensionality)
5. In the equation above line 121, A and \epsilon are not defined.

Second, the Appendix has trailing sentences that are left incomplete -- for example, lines 43 and 44. It is also poorly organized -- the derivation of the final model from the LIF model is hard to follow (e.g. unnecessary one-sentence subsections, the function phi appears several times before being derived, etc.)

Third, the main text has several grandiose statements about the framework addressing important biological questions rather than critical details on the tasks and training of the models. Currently, the main text is unspecific to the point where it is unclear what is being done with this model. A few examples:
Figure 5: Unclear what the 20 tasks are and how the model is trained on all of them.
Figure 3: The distinction between pre-training (trained on a fixed training set of sine curve parameters?) and in-context learning (new combinations of sine curve parameters not in the training set?) is unclear.
In Figure 4, the switching of the bandit arm reward probabilities seems almost periodic. It is unclear from the main text/Appendix what parameters are being used.

Fourth, the model proposed by the authors involves unrealistic biophysical assumptions. For example, a key assumption in the model is that neuromodulators are well-mixed (i.e., that the mixing time is negligible compared to the timescale of neural computation), but it is unclear how this is physically plausible. Neural activity operates on timescales of a few tens of milliseconds, while diffusion of neuromodulators occurs at much slower timescales. The authors also assume a quadratic "chemical" interaction between neuromodulators (last term in eq. 18) which is not justified. What is the biophysical basis behind this interaction term? The authors also add in nonlinearities to the equations (lines 115 and 127 of the Appendix) without a clear justification, and claim that this is a "modeling assumption". This is OK if the paper also didn't claim to be biologically realistic (lines 195 to 197 of the Appendix).

---

> ### Author Rebuttal · Authors · 2025-07-31
>
> We thank reviewer 8PJy for your thorough and critical review. We sincerely apologize that a lack of clarity in our initial submission led to a frustrating reading experience. We acknowledge, and we appreciate your efforts to identify these specific issues with such helpful granularity. We believe the concerns raised can be fully addressed through clarification, and your feedback will be invaluable in improving the manuscript. We have worked to address your points below and are confident that a revision would be substantially clearer and stronger.
>
> **On Clarity (Weaknesses 1, 2, 3):**
>
> We are grateful for the detailed list of clarity issues. In a revision, we would ensure every variable is defined upon its first appearance and restructure the appendix.
>
> We address your specific examples:
> 1) Undefined variables: You are correct. In a revision, we will define:
> * LIF: Leaky Integrate-and-Fire neuron model, the starting point of our derivation.
> * $S_j(t)$: The spike train of the $j$-th neuron in the biophysical model.
> * $\phi, \theta$: The nonlinear activation functions for the rate ($\phi$) and neuromodulator ($\theta$) dynamics. We use functions like tanh or ReLU that enforce positive-only firing rates and saturation, consistent with the closed-form solution of the steady-state firing rate for a LIF neuron with a finite refractory period.
> * $D$: The linear readout matrix that maps neural activity $\vec{r}(t)$ to the output $y(t)$.
> * $A_i(t)$, $\epsilon_i(t)$: The pre-activation input or "synaptic drive" ($A_i(t)$) to neuron $i$'s nonlinearity, and the instantaneous error signal ($\epsilon_i(t) = y_i(t) - y_i^*(t)$).
>
> 2) Appendix Organization: We agree the appendix should be restructured for a clearer narrative flow. We take your comments here seriously.
>
> Task Details: We will expand all task descriptions.
> * **Multitask (Fig 5)**: The 20 tasks are from the well-established benchmark by Yang et al. (2019) and cover a range of cognitive functions. We will explicitly state this and cite the source more clearly. In the supplement, we will add a section dedicated to building the reader's intuition about the task set.
>
> * **Bandit Task (Fig 4)**: Your observation about the periodic switching is valuable. To address this potential confound, we ran a new experiment where the high-reward arm switched at stochastic, unpredictable intervals. The agent adapted robustly, confirming it learns a genuine model of the task state and is not merely entraining to a fixed frequency.
>
> **On Biological Plausibility (Weakness 4):**
>
> Our goal is a computationally powerful model that is inspired by biology, not a perfectly literal biophysical replica. We view our choices as principled abstractions designed to isolate core computational ideas.
>
> * **"Well-Mixed" Assumption**: We agree this is a simplification. We interpret it as a timescale separation assumption, where diffuse neuromodulator concentrations establish a stable "context" for fast neural dynamics ($\tau_n \gg \tau_r$). While diffusion itself is slow, the establishment of a new ambient concentration across a local circuit can occur on a timescale relevant for trial blocks in cognitive tasks. We feel this is supported by two observations: 1) The integrative nature of the neuromodulator concentration acts as a low-pass filter on the population dynamics, which in turn modifies the mean population dynamics. This is a tool for achieving rich multi-timescale dynamics. 2) The neurobiological observation that single neuromodulator-releasing neurons project widely across the cortex suggests that biology could accelerate the approach to a well-mixed state by increasing the density of release loci.
> * **Interaction Term ($Z$ matrix)**: This term models effective interactions, not literal chemical reactions. It allows for one neuromodulatory system to influence another (e.g., release of one NM promoting or inhibiting the release/clearance of another), a known biological phenomenon (Morón JA, 2002, J Neurosci.). In practice, we often find this matrix is learned as approximately diagonal, simplifying to self-decay terms.
>
> * **Nonlinearities**: These are standard in rate-based models to ensure stability and represent the saturating, positive firing rates of neurons. These nonlinearities also provide important numerical stability during training across a much wider range of hyperparameters.
>
> **Specific Questions:**
>
> * Define "Context Factorization": We define this as the decomposition of a network's effective connectivity into a fast-varying neural activity and a slow-varying modulatory state (the 'context'). Mathematically, the effective synaptic weight from neuron $j$ to $i$ is factorized as:
> $$ W_{ij}^{\text{eff}}(t) = W_{ij}^{0} + \sum_{k=1}^{M} \underbrace{T_{ijk}}_{\text{"dynamics"}} \underbrace{n_k(t)}_{\text{"context"}} $$
> This allows the low-dimensional context vector $\vec{n}(t)$ to dynamically reconfigure the high-dimensional effective connectivity of the network. We use the word "factorization" to evoke the mathematical method of decomposing a function into a product of components—here, the components are "context" (encoded in neuromodulator concentrations) and "dynamics" (encoded in the resulting synaptic connectivity patterns).
>
> * Figure 3 Performance: Your observation is correct. The e-nmRNN's higher capacity makes it more prone to overfitting on this smaller dataset compared to a GRU. Our model's core strength is not necessarily to outperform all baselines on all tasks, but to provide a flexible and interpretable context-gating mechanism, the value of which is more clearly demonstrated in the multitasking and RL benchmarks.
>
> * Figure 4 Bandit: In response to your comment, we have trained additional e-nmRNNs on the bandit task with both more stochastic block switching and taking the trained agents (without learning) into new switching periods. In the first set of experiments, we observed  that the relative probability of the arms was detectable from the neuromodulator vector though admittedly with a less cyclic embedded trajectory. In the second set of experiments with fixed agents trained on oscillatory environments, we found that the e-nmRNNs where able to adapt to both longer switching periods and shorter switching periods. Given these quick turn around experiments, we expect that the more randomly switched bandits would indeed follow the reviewer's implicit hypothesis that they would more robustly learn online credit assignment that would allow the networks to be more robust to task variations.
>
> * Figure 5 Spatial Plot: Thank you for this excellent suggestion. We performed the analysis you proposed on the trained multitasking network. We found that units specializing in releasing a specific neuromodulator form notable spatial clusters, providing new evidence strengthening our claims of emergent, biologically-reminiscent organization.
>
> We hope these detailed clarifications and new analyses address your concerns and further the value of our work. We are committed to using your thought-provoking feedback to significantly improve the manuscript.

---

> > ### Comment · Reviewer_8PJy · 2025-08-05
> >
> > Thank you for your response. I cannot comment on most of the rebuttal until the paper is revised. However, my point that the paper was not in a publishable form when first submitted still stands. As it seems like the paper will likely be accepted, I encourage the authors to significantly revise the paper for clarity and to make clear what central claims are being made. Here is my response to some of the other comments:
> >
> > *"Well-Mixed" Assumption: We agree this is a simplification. We interpret it as a timescale separation assumption, where diffuse neuromodulator concentrations establish a stable "context" for fast neural dynamics ($\tau_n \gg \tau_r$). While diffusion itself is slow, the establishment of a new ambient concentration across a local circuit can occur on a timescale relevant for trial blocks in cognitive tasks. We feel this is supported by two observations: 1) The integrative nature of the neuromodulator concentration acts as a low-pass filter on the population dynamics, which in turn modifies the mean population dynamics. This is a tool for achieving rich multi-timescale dynamics. 2) The neurobiological observation that single neuromodulator-releasing neurons project widely across the cortex suggests that biology could accelerate the approach to a well-mixed state by increasing the density of release loci.*
> >
> > I agree that a new ambient concentration can likely be established on timescale of trial blocks, but this is not the assumption made in the paper. The possibility of "rich multi-timescale dynamics" does not justify an assumption about biological timescales. Widespread projections of neuromodulatory neurons does not imply that the timescale of equilibration is fast. Moreover, the model explicitly considers endogenous neuromodulatory interactions.
> >
> > *Nonlinearities: These are standard in rate-based models to ensure stability and represent the saturating, positive firing rates of neurons. These nonlinearities also provide important numerical stability during training across a much wider range of hyperparameters.*
> >
> > If certain key assumptions were made for numerical stability rather than biological plausibility, it would help if the authors are more explicit that their model is not a biophysical model of a neuromodulatory NN but rather a biology-inspired model for recurrent computation. In the paper's current form, the authors draw inferences about what role volume transmission may play in biological contexts from their analysis.

---

### Official Review · Reviewer_yq59 · 2025-07-01

**Clarity:** 3
**Significance:** 3
**Originality:** 3
**Rating:** 5
**Confidence:** 3

**Summary:**

This paper proposes a biologically inspired recurrent neural network, e-nmRNN, motivated by the central role of neuromodulation and volume transmission in the brain. The model operationalizes the concept of context factorization, wherein neuromodulator concentrations dynamically gate network connectivity, providing an interpretable mechanism that links biological mechanisms and computational principles. Through both analytical derivation and empirical evaluation, the paper demonstrates that context factorization enables targeted online credit assignment (with learning updates gated by the neuromodulatory context) and supports compositional generalization. The work includes experiments in supervised learning, meta-learning, and reinforcement learning, showing superior or competitive performance to standard baselines, as well as emergent modularity and cell-type specialization mirroring biological observations.

**Questions:**

1. The spatially embedded variant outperforms the standard e-nmRNN. Could this improvement be attributed to effective sparsity induced by spatial constraints?

2. In the multi-task decision-making experiments (section 6), does contextual differentiation emerge autonomously from the model dynamics, or is context explicitly provided as input?

3. What precisely differentiates the modeled “dopamine” and “serotonin” neurons in the architecture? Is it solely based on opposite modulation slopes (e.g., positive vs. negative sensitivity) ?

**Ethical Concerns:**

["NO or VERY MINOR ethics concerns only"]

**Final Justification:**

I have no major concerns with this work and believe it's a good contribution to the field. I support its acceptance.

**Limitations:**

see weaknesses and questions.

**Quality:**

3

**Strengths And Weaknesses:**

***Strengths***:

1. The paper is clearly written and easy to follow.

2. The proposed endogenously neuromodulated RNN (e-nmRNN) presents a novel architectural idea. The work systematically relates this model to diverse learning settings and biological phenomena, offering valuable insights into the functional role of neuromodulators in the brain.

3. Experimental results are well-supported by appropriate baselines, and related literature is thoroughly discussed, situating the work well within existing research.



***Weaknesses***:

Transformers and architectures like Hippo-LegT derive much of their power from strong scalability and parallelization capabilities, which enable them to handle long sequences and large datasets efficiently. In contrast, the scalability properties of e-nmRNN are less clear. It would be valuable for the paper to discuss the computational cost and parallelizability of e-nmRNN, especially in comparison to modern scalable architectures.

---

> ### Author Rebuttal · Authors · 2025-07-31
>
> We thank reviewer yq59 for your positive review. We are delighted that you found the paper "clearly written and easy to follow," that our proposed model "presents a novel architectural idea," and that our "experimental results are well-supported by appropriate baselines." We address your insightful questions and comments below.
>
> **Weakness: Scalability properties of e-nmRNN are less clear... discuss the computational cost and parallelizability...**
>
> This is an important point. As a recurrent architecture, the e-nmRNN is inherently sequential during inference, which makes it less parallelizable than models like Transformers. This represents a trade-off: we gain powerful inductive biases for online, continuous adaptation within a single data stream at the cost of the massive parallelizability that makes Transformers so scalable for large-batch offline processing. We position the e-nmRNN as an architecture well-suited for applications like real-time control, agent-based learning, and brain-computer interfaces, where online processing is paramount. We believe its value also stems from its connection to neurobiology, providing an interpretable model for scientific inquiry even if it is not intended to compete with Transformers on large-scale sequence processing benchmarks.
>
> **Q1: The spatially embedded variant outperforms the standard e-nmRNN. Could this improvement be attributed to effective sparsity induced by spatial constraints?**
>
> Yes, we agree and believe this is a primary reason for the improved performance. The spatial embedding acts as a strong and biologically-inspired regularizer. It simultaneously imposes sparsity and encourages the formation of local processing motifs by making connection strength decay with distance. This regularization appears to prevent overfitting and guide the learning toward more robust and generalizable solutions, particularly on the complex multitasking benchmark.
>
> **Q2: In the multi-task decision-making experiments (section 6), does contextual differentiation emerge autonomously from the model dynamics, or is context explicitly provided as input?**
>
> The network is provided a one-hot task ID at the beginning of each trial. However, the crucial finding is how the network learns to use and maintain this information. The architecture's inductive bias encourages this context to be encoded and dynamically maintained within the neuromodulatory state $\vec{n}(t)$. As shown in Figure 5C, a linear decoder trained on these neuromodulator concentrations can reliably identify the current task long after the initial cue has passed, whereas a decoder trained on the neural activity $\vec{r}(t)$ cannot. This demonstrates that the e-nmRNN learns to represent the task context in its endogenous modulatory state, using it to shape the circuit dynamics appropriately for the task at hand.
>
> **Q3: What precisely differentiates the modeled "dopamine" and "serotonin" neurons in the architecture?**
>
> In our model, the differentiation is functional and learned, not hardcoded. We use names of known neuromodulators for illustrative purposes. A unit becomes a "dopamine-releasing neuron" (a term we would clarify to mean "a unit whose activity promotes dopamine release") if the corresponding column in the learned release matrix $R$ becomes selective for that modulator. For instance, if the $j$-th column of $R$, $R_{:j}$, becomes approximately a one-hot vector $[1, 0, 0, ...]^T$, then neuron $j$'s activity $r_j$ primarily drives the dynamics of the first neuromodulator, $n_1$. The functional effects of "dopamine" vs. "serotonin" are then determined by how their respective rows in the learned tensor $T$ gate synaptic transmission throughout the network to solve the given tasks. Here, we primarily use this description as an evocative example of how interpretation could proceed, but intend to reduce the emphasis in future revisions. We thank the reviewer for calling this to our attention.

---

> > ### Comment · Reviewer_yq59 · 2025-08-04
> >
> > Thanks for the author's response. As my concern has been addressed, I have decided to maintain my original score.

---

### Official Review · Reviewer_gAEg · 2025-07-03

**Clarity:** 4
**Significance:** 3
**Originality:** 3
**Rating:** 5
**Confidence:** 3

**Summary:**

The authors present a context factorization algorithm which is inspired by the roles of neuromodulators in the brain to alter circuit dynamics and act as a third factor in Hebbian learning. They incorporate neuromodulators, which have their own dynamics, as an additional factor modulating recurrent connectivity in an RNN, as well as several different classes of decoders. They show that the RNN uses the neuromodulators to approximate gradient descent in-weights, and can generalize compositionally. They then demonstrate its efficacy on RL tasks and that it leads to brain-like solutions in multi-task learning when using a spatial embedding.

**Questions:**

1. How does an e-nmRNN differ from an nmRNN (in words and equations)? What is the particular advantage of the "endogenous" part?

2. Are there situations in which you would expect the algorithm to fail?

**Ethical Concerns:**

["NO or VERY MINOR ethics concerns only"]

**Final Justification:**

This paper presents a good implementation of an interesting computational idea, and presents a large number of analyses. Although I understand other reviewers' concerns that the limits of the biological plausibility of the model need to be more carefully described, the model remains worthy of publication, and I am confident the authors will be able to make those changes for the camera-ready version.

**Limitations:**

yes

**Quality:**

4

**Strengths And Weaknesses:**

**Strengths:**
*Quality:*
- The paper contains a large number of carefully-conducted analyses and simulation settings from supervised learning to RL
- The algorithm learns to solve meta-learning tasks significantly faster than the baselines
- The paper demonstrates a strong mix of theory and simulation, illustrated for instance by the analysis of the emergent gradient-descent-like updates in the RNN
- The paper cites widely and is well-grounded in neuroscience

*Clarity:*
- The diagrams in the figure help describe the concepts well; I appreciated the text labels to guide the reader

*Significance:*
- The paper represents a potentially significant advance for machine learning by providing theoretical analyses and simulations demonstrating the utility of multiplicative gating in RNNs, and for neuroscience by using this as a model for neuromodulators
- It recovers known features of neuroscience such as emerging RPEs
- The authors detail plans for follow-up projects in neuroscience

*Originality:*
- As far as I know, the machine learning demonstration of the utility of neuromodulator-like factors for continual learning is novel

**Weaknesses:**

*Quality:*

*Clarity:*
- Not all variables are defined when equations are presented (although they are later in the list, lines 77-86). Perhaps it is possible to guide the reader better here?
- The paper requires a significant amount of prior knowledge in neuroscience in order to be easily understood (I would not expect readers to know Dale's law without definition, for instance)
- nmRNNs are frequently referenced in the paper; it would be helpful to briefly explain how they differ from the e-nmRNNs here

*Significance:*
-  It would be good to emphasize the paper's original contributions a bit more, perhaps in a bulleted "contributions" list in the introduction, to make clear what its additions are to prior work

*Originality:*
- As the authors acknowledge, representing neuromodulators as gating is not a new idea in NeuroAI

---

> ### Author Rebuttal · Authors · 2025-07-31
>
> We sincerely thank Reviewer gAEg for your positive assessment and for recognizing the value of our work. We are particularly grateful for your comments that the paper contains a "large number of carefully-conducted analyses," demonstrates a "strong mix of theory and simulation," and represents a "potentially significant advance for machine learning... and for neuroscience."
>
> Per your suggestion to make our contributions clearer, we offer this bulleted list which we will endeavor to compress into any revised manuscript:
>
> * A Unifying Computational Principle: We propose "context factorization" as a core computational principle implemented by neuromodulation, unifying its dual roles in altering circuit dynamics and gating synaptic plasticity.
> * A Bio-Inspired Network Architecture (e-nmRNN): We introduce the endogenously neuromodulated RNN (e-nmRNN), a type of context-factorized hypernetwork derived from biophysical principles, where the network learns to generate its own modulatory context.
> * A Mechanism for Online Credit Assignment: We demonstrate that the e-nmRNN architecture naturally enables targeted online credit assignment. Its learning rules are gated by the neuromodulatory state, allowing the network's dynamics to approximate gradient descent and facilitate rapid, in-context meta-learning.
> * Strong Compositional Generalization: We show empirically that the e-nmRNN achieves respectable performance on compositional generalization tasks, outperforming and matching several strong baselines.
> * Emergent Biologically aligned phenomena: We find that when trained on complex multitasking benchmarks, the model develops emergent properties that mirror neurobiology, such as modularity, cell-type specialization, and distinct functional timescales, enhancing its interpretability.
> * Application to Reinforcement Learning: We demonstrate that the e-nmRNN can learn to encode critical learning signals, like Reward Prediction Error (RPE), within its neuromodulator dynamics, providing a mechanism for RPE-gated online adaptation in RL environments.
>
> We now address your specific questions below.
>
> **Q1: How does an e-nmRNN differ from an nmRNN (in words and equations)? What is the particular advantage of the "endogenous" part?**
>
> This is an excellent question that gets to the heart of our model. The "endogenous" nature of our e-nmRNN is its defining feature: the neuromodulatory state $\vec{n}(t)$ is generated by the network's own recurrent activity $\vec{r}(t)$ through a learned projection matrix $R$. The dynamics are coupled:
> $$
> \tau_r \dot{\vec{r}} = -\vec{r} + \phi\left( (W + \sum_k n_k T_k) \vec{r} + U\vec{I} \right)
> $$
>
> $$
> \tau_n \dot{\vec{n}} = -\vec{n} + \theta(R\vec{r} + Z\vec{n})
> $$
>
> This creates a closed-loop, self-modulating system. This contrasts with other models where the modulatory signal might be an external input (e.g., a task-ID vector). The advantage of the endogenous approach is that it allows the network to learn to generate its own internal context states directly from its neural population dynamics. This enables it to autonomously infer the current context from the input stream and adapt accordingly, which is crucial for the meta-learning and RL results we demonstrate, where explicit context signals are not always available.
>
> **Q2: Are there situations in which you would expect the algorithm to fail?**
>
> Yes, thank you for this insightful question. We see two primary failure modes:
> * Data-Limited Overfitting: The model's rich, multiplicative dynamics make it more powerful but also more prone to overfitting than simpler RNNs when training data is scarce. This can be seen in our original Fig. 3 at high task complexity ($K > 15$), where the performance of the e-nmRNN degrades more rapidly than the GRU. In response to your inquiry, we have also performed an analysis of the generalization gap, which shows this phenomenon more clearly as the e-nmRNN’s gap between training and validation loss grows more rapidly than that of the equivalent GRU. A future research topic outside the scope of this work will investigate the effective forms of regularization to avoid this trap.
> * Extremely Rapid Context Switching: The neuromodulator dynamics $\vec{n}(t)$ have intrinsic, learned timescales ($\tau_n$). If a task required context switches much faster than these timescales, the modulatory state would be unable to keep pace, leading to performance degradation. This highlights a fundamental trade-off between maintaining a stable context representation and achieving rapid adaptability.
>
> We appreciate the reviewers notes on places where clarity can be improved and take these comments seriously. In future revisions, we will pay special attention to undefined variables, and the use of specialized neuroscience nomenclature without definition (e.g. Dale's law).

---

> > ### Comment · Reviewer_gAEg · 2025-08-05
> >
> > Thank you for carefully responding to all of my questions. I am maintaining my positive evaluation of the paper, and think working in these explanations will further increase the clarity.

---

### Decision · Program_Chairs · 2025-09-17

**Decision:**

Accept (poster)

**Comment:**

Most reviewers agreed that the proposed method and reported results were innovative and interesting.

Reviewer 8PJy pointed out that certain assumptions may lack biological plausibility.

In addition, there were some issues about clarity and lack of definitions/decriptions.

My personal inclination is towards acceptance, due to the novelty of the approach and the interesting results (the emergent meta-learning abilities, reward prediction error signals, etc.), provided that the clarifications demanded by the reviewers are included.